# Functional role of the Frizzled linker domain in the Wnt signaling pathway

Seung-Bum Ko[1], Emiko Mihara [2], Yedarm Park[1], Kyeonghwan Roh[1], Chanhee Kang [1], Junichi Takagi [2], Injin Bang [1,3✉] & Hee-Jung Choi [1✉]

The Wnt signaling pathway plays a critical role in the developmental and physiological processes of metazoans. We previously reported that the Frizzled4 (FZD4) linker domain plays an important role in Norrin binding and signaling. However, the question remains whether the FZD linker contributes to Wnt signaling in general. Here, we show that the FZD linker is involved in Wnt binding and affects downstream Wnt signaling. A FZD4 chimera, in which the linker was swapped with that of the non-canonical receptor FZD6, impairs the binding with WNT3A and suppresses the recruitment of LRP6 and Disheveled, resulting in reduced canonical signaling. A similar effect was observed for non-canonical signaling. A FZD6 chimera containing the FZD1 linker showed reduced WNT5A binding and impaired signaling in ERK, JNK, and AKT mediated pathways. Altogether, our results suggest that the FZD linker plays an important role in specific Wnt binding and intracellular Wnt signaling.

[1] Department of Biological Sciences, Seoul National University, Seoul 08826, Republic of Korea. [2] Laboratory for Protein Synthesis and Expression, Institute for Protein Research, Osaka University, Suita, Osaka 565-0871, Japan. [3] Present address: Laura and Isaac Perlmutter Cancer Center, New York University Langone Medical Center, New York 10016 NY, USA. ✉email: injin.bang@nyulangone.org; choihj@snu.ac.kr

The Frizzled receptor (FZD) belongs to the class F subfamily of G protein-coupled receptors. There are 10 FZD subtypes encoded in the human genome, and these are well known as the main receptors for Wnt ligands[1]. On the cell surface, FZD recognizes Wnt ligands through a cysteine-rich domain (CRD); together with essential co-receptors, it activates specific downstream signaling depending on the FZD and Wnt subtypes[2]. Wnt signaling is traditionally categorized into two broad pathways, namely canonical, also known as the β-catenin dependent pathway, and noncanonical, also known as the β-catenin independent pathway[3,4]. Canonical signaling requires the LRP5/6 co-receptor, whereas noncanonical signaling is more complicated as the required co-receptors vary for different noncanonical pathways. Once the specific downstream signal is initiated, it is further transduced on the cytosolic side of the cell through a scaffold protein called Disheveled (DVL)[5].

CRD is the orthosteric binding site for Wnt, and the molecular details of its interaction with Wnt ligands and an atypical ligand called Norrin have been published[6–9]. However, the detached structures of CRD-ligand complexes are not sufficient to elucidate how FZD conformationally accommodates ligand binding and initiates a signal across the cell membrane. Computational modeling was attempted to understand how the ligand-binding domain is linked to the transmembrane domain of FZD[10,11]. However, there is still more to learn regarding the connectivity between FZD-ligand interactions and downstream signaling. We have previously demonstrated that the linker domain of FZD4 contributes to the interaction with Norrin and downstream signaling[10], which is yet to be structurally elucidated. Although Norrin can activate canonical Wnt signaling, it is structurally different from Wnt. Norrin forms a cystine-knot homodimer linked by intermolecular disulfide bonds and binds the FZD4 CRD in 2:2 stoichiometry, through a single interface; this interface overlaps the Wnt binding site. In addition to this shared interface, Wnt has another interaction site through which Wnt's highly conserved acylated lipid group is accommodated. It has been clearly shown in two published structures that Wnt can bind to the CRD in either 1:1 or 2:2 stoichiometry[6,8]. Norrin and Wnt also differ in selectivity for FZD subtype; Norrin binds specifically to FZD4 only[12], while Wnts have been shown to interact with one or multiple FZD subtypes[13–15]. For example, WNT8B only activates FZD2, whereas WNT3A activates canonical signaling through a wide range of FZD subtypes, namely FZD1, 2, 4, 5, 7, 8, 9, and 10[16–18].

The structures of Xenopus Wnt8 (XWnt8) in complex with the mouse FZD8 CRD[6] and human WNT3 in complex with the mouse FZD8 CRD[8] showed that Wnt and CRD residues constituting the binding interfaces are mostly conserved, with some variations. As the linker domain is the most sequentially varied region among FZDs[19], we speculated that this linker region, in coordination with the CRD, would contribute to the selectivity of FZD for the Wnt ligands. We aimed to evaluate the contribution of the FZD linker domain to Wnt-FZD interaction, FZD oligomerization, and Wnt signaling using the chimeric mutants of FZD4 and FZD6. The former activates canonical signaling in response to WNT1, WNT3A, and WNT5A[20,21], and the latter activates noncanonical signaling in response to WNT5A[22]. Using cell surface binding assays, we found that the linker domain is important for the subtype-specific interaction of the FZD with the Wnt ligand, thus affecting downstream signaling. Using bioluminescence resonance energy transfer (BRET) assays, we also demonstrated that linker-swapped mutants with reduced canonical signaling were defective in LRP6 and DVL2 recruitment. In addition, the linker domain of FZD6 was shown to be involved in the activation of noncanonical signaling. Altogether, our results show that the FZD linker domain is involved in multiple aspects of signal activation, i.e., from specific Wnt binding to selective downstream Wnt signaling.

## Results

### Linker-swapped mutants were designed without disrupting conserved cysteine residues

To investigate whether the FZD linker domain is involved in ligand binding and hence, in downstream Wnt signaling, we designed a FZD4 construct lacking the linker region (FZD4Δlinker) and several FZD4 chimeric mutants, where the CRD, linker, or both (CRDlinker), were replaced with corresponding regions of FZD3, FZD5, and FZD6 (Fig. 1a, Supplementary Table 1). FZD4 and FZD5 were chosen as representative canonical Wnt receptors, and FZD3 and FZD6, which have been shown to activate noncanonical signaling in response to WNT5A[22,23], were chosen as case studies for noncanonical signaling in response to WNT5A.

Sequence alignment of the linker domains of FZD3, FZD4, FZD5, and FZD6 showed that there are three highly conserved Cys residues in the linker domain (C181, C200, and C204 in FZD4). In the crystal structure of the N-terminal domain deletion mutant of FZD4, C181 and C200 form an intra-linker disulfide bond whereas the remaining C204 forms a disulfide bond with C282 in extracellular loop 2, which is also conserved across all FZD subtypes and even in Smoothened[24]. Mutations of one of these conserved Cys residues in FZD6 have been shown to affect the expression of FZD6 on the surface of the plasma membrane[25]. As we swapped the FZD4 linker region of 161–203 with the corresponding region of other FZD subtypes, we ensured that conserved cysteines were present in all mutants, keeping the two disulfide bonds, one within the linker and another between the linker and extracellular loop 2, intact (Supplementary Fig. 1). A recent publication by Tsutsumi et al. argued that FZD5/8 has a special CXC motif in extracellular loop 3, with cysteines that are likely to form disulfide bonds with two 'free' cysteines in the linker region[26]. Under this assumption, the FZD4 chimera, containing the FZD5 linker (FZD4_5linker), would end up misfolding owing to the disruption of the disulfide network. However, our immunofluorescence assay (IFA) clearly showed that the FZD4_5linker was trafficked to the cell surface (Fig. 1b, Supplementary Fig. 2). Furthermore, its expression level on the surface, as measured by flow cytometry analysis, was comparable to that of FZD4 (Supplementary Fig. 3). Based on these results, additional CRD-, linker-, or CRDlinker-swapped mutants of FZD4 and FZD6 (Supplementary Table 1) were designed to investigate the significance of the linker domain in Wnt signaling. The surface expression and expression levels of all mutants were confirmed using IFA, flow cytometry analysis, and enzyme-linked immunosorbent assay (ELISA) (Supplementary Figs. 2–4).

### FZD linker domain is actively implicated in Wnt signaling

Wnt signaling is often upregulated in cancer, suggesting that commonly used cancer cell lines may have fully functional endogenous FZDs as well as constitutively secreted Wnts. To eliminate the interference of endogenous FZDs, all our cell-based canonical signaling assays were performed using the FZD null cell line (dFZD1-10$^{-/-}$ HEK293T cells)[27]. Before we examined the effect of the FZD linker on canonical Wnt signaling, we first investigated the importance of the CRD in the canonical signaling pathway. WNT1 and WNT3A were applied to cells transfected with FZD4 CRD-swapped mutants as well as the wild-type FZD3, FZD4, FZD5, and FZD6. TOPFlash assays showed that FZD4 and FZD5 responded to WNT1 and WNT3A, while FZD3 and FZD6 remained silent, which was as expected (Fig. 2a). Replacing the FZD4 CRD with the CRD of FZD3 or FZD6 was shown to completely eliminate the activity of WNT1 and WNT3A (Fig. 2a), suggesting that the specific Wnt-CRD interaction is an important determinant of downstream signaling pathways.

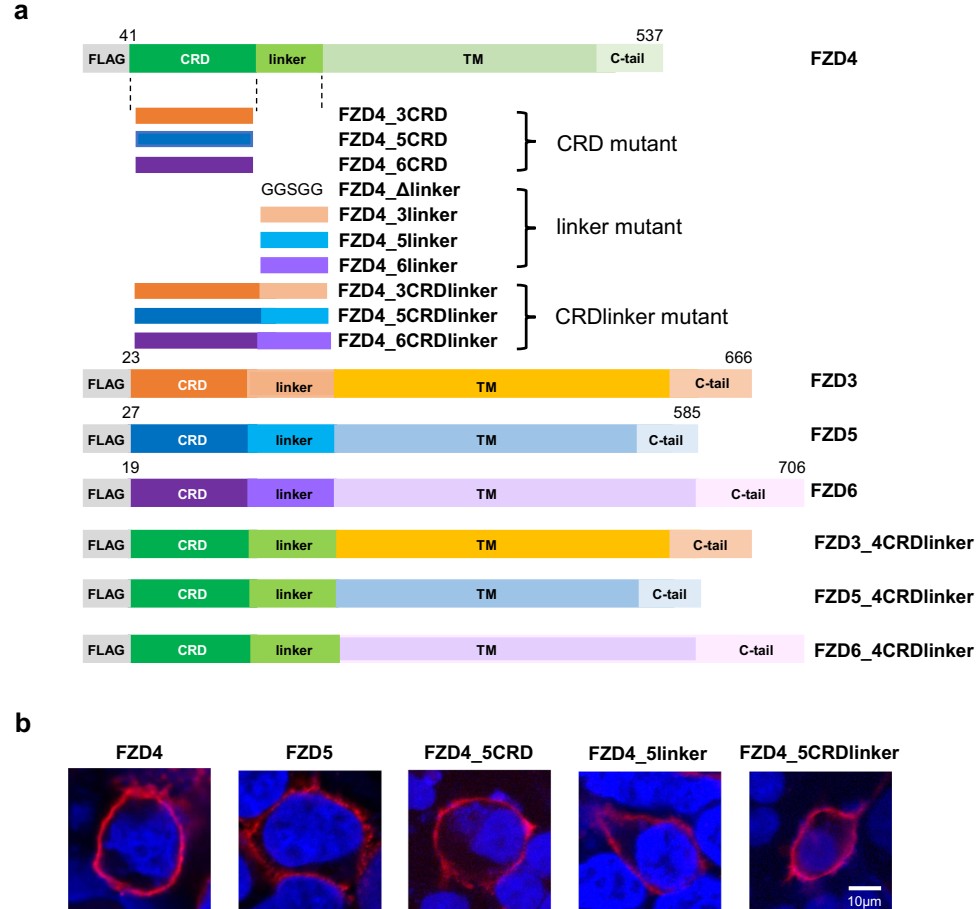

**Fig. 1 Design of FZD mutant constructs and observation of surface expression. a** Schematic representation of wild-type and various mutant constructs of FZD3, 4, 5, and 6. All constructs carry an N-terminus Flag tag. Numbers indicate amino acid numbering in the indicated FZD subtype. A complete list of FZD mutant constructs is given in Supplementary Table 1. **b** Surface expression of representative FZD4 mutant constructs was detected with anti-FLAG antibody (red) and nuclei was stained with Hoechst33342 (blue) (Scale bars: 10 μm). Surface expression of various FZD mutants was analyzed with immunofluorescence assay, flow cytometry analysis, and surface ELISA assay (Supplementary Figs. 2–4).

The impact of FZD linkers on canonical Wnt signaling was then examined using TOPFlash assays with FZD4 linker-swapped mutants and FZD4Δlinker, where the linker domain was deleted and the CRD was linked to TMD with the GS linker (Fig. 1a, Supplementary Table 1). FZD4Δlinker showed reduced activity compared to the wild-type (Fig. 2b), confirming that, as it does for Norrin, the linker domain plays a role in the signaling of the canonical Wnt ligand. Swapping the FZD4 linker with the FZD5 linker (FZD4_5linker) had almost no effect on the ability of FZD4 to respond to both WNT1 and WNT3A. The retained activity of this chimera compared to the Δlinker mutant is in accordance with the fact that the FZD5 subtype can also activate canonical signaling in response to WNT1 and WNT3A. Conversely, when we swapped the FZD4 linker with the corresponding linker region of the noncanonical FZDs, FZD3 and FZD6 (FZD4_3linker and FZD4_6linker, respectively), we observed a dramatic decrease in canonical signaling activity even though these mutants contained the intact FZD4 CRD (Fig. 2b). It is noteworthy that the effect of the FZD4 linker on the canonical signaling was more evident when recombinant WNT3A protein was treated instead of WNT3A plasmid transfection, probably because plasmid transfection may result in a longer and more excessive exposure to WNT3A than that of protein treatment (Fig. 2b).

Therefore, although the linker-swapped mutants, FZD4_3linker and FZD4_6linker, cannot completely eradicate the canonical Wnt

activity as shown in the CRD-swapped mutants, FZD4_3CRD and FZD4_6CRD, the linker is certainly important for FZD to have its maximal signaling capacity, suggesting that the linker assists the CRD in Wnt recognition.

**FZD linker affects the binding to the Wnt ligand.** Following the TOPFlash assay, we wanted to investigate whether there was a direct correlation between the reduced canonical signaling activity of the linker-swapped mutants and their binding affinity towards the ligand. Unfortunately, affinity measurement between WNT3A and FZD in a purified state is challenging because of the hydrophobicity of both proteins, requiring detergent for their solubility, which could interfere with their binding[28,29]. Given this, an immunostaining assay using an alkaline phosphatase (AP)-conjugated antibody, which was demonstrated as an alternative technique that can be used to detect the Wnt ligand-binding to FZDs overexpressed on the cell surface[13,30,31], could be used. We applied this method to our system by detecting FZD-bound WNT3A with AP-conjugated anti-WNT3A antibodies and visualizing the AP tags with an NBT/BCIP substrate. More intense staining would imply more WNT3A bound to the cells, indicating a stronger binding. To confirm that our staining protocol worked and that the target FZD constructs were expressed properly, we first performed staining with an anti-FLAG antibody on the N-terminal FLAG tag of FZD, and we observed uniform expression levels at the cell membrane (Fig. 3).

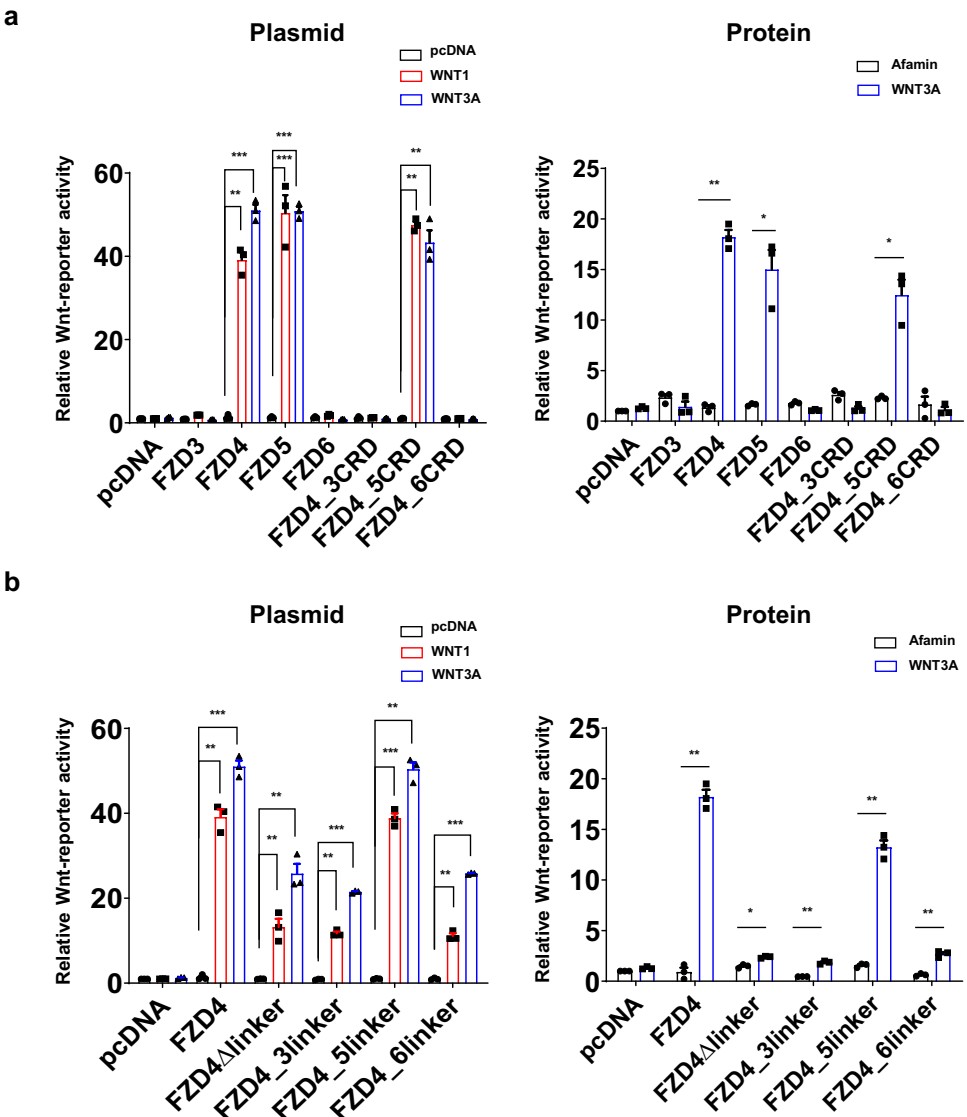

**Fig. 2 Canonical signal activity of FZD linker swap mutants.** The effects of various FZD4 mutants on WNT1 and WNT3A signaling were observed using TOPFlash assays. The bar graphs show the results of plasmid transfection (left) and protein treatment (right) for (**a**) CRD-swapped mutants and (**b**) linker-swapped mutants of FZD4. The error bar indicates the SEM (standard error of mean) of three independent experiments. Statistical comparisons were performed using the two-tailed $t$-test. '***' represents $P < 0.001$, '**' represents $P < 0.01$, and '*' represents $P < 0.05$ for TOPFlash assays on various FZD4 CRD and linker-swapped mutants.

Then, cells transfected with each FZD construct were treated with conditioned media producing WNT3A, followed by PBS washing, and bound WNT3A was stained with an anti-WNT3A antibody. As shown in Fig. 3, the strong intensity was observed in cells transfected with FZD4, demonstrating that WNT3A efficiently bound to wild-type FZD4. FZD4 linker mutants that showed a lesser response to WNT3A displayed less intense staining. The FZD4 CRD chimeras, FZD4_3CRD and FZD4_6CRD that did not exhibit canonical activity, did not show measurable AP-staining above background (Supplementary Fig. 5). In contrast, the FZD4_5linker and FZD4_5CRD mutants showed intensity comparable to that of wild-type FZD4 (Fig. 3, Supplementary Fig. 5). These results indicate that the linker swap affected canonical signaling activity by altering the interaction between FZD and the Wnt ligand. This result is reminiscent of the importance of the FZD4 linker in Norrin binding, as it has been shown that the presence of the FZD4 linker in accompaniment to CRD improved the binding affinity for Norrin by ~10 fold.

Oligomerization of FZD is a common and often critical step in signal amplification[32,33]. It has been shown that Wnt binding induces FZD oligomerization[34]. The structure of the Wnt-bound FZD CRD was solved in both 1:1 and 2:2 stoichiometry, suggesting that Wnt can link two FZDs together at the extracellular level. We wondered if the linker domain also affects the ligand-induced oligomerization of FZD, as we observed that the linker domain influenced the binding to Wnt. We performed BRET analyses to observe Wnt-induced homo-oligomer formation. The saturating ratio of BRET acceptor/donor was determined prior to the assays (Supplementary Fig. 6). In the absence of Wnt, there was no difference in oligomerization tendency among the different FZD constructs (Supplementary Fig. 7). Upon treatment with WNT3A, FZD4 showed the greatest increase in ΔBRET signal followed by FZD4_5linker (Fig. 4a), which is consistent with the AP-stained data showing interaction with WNT3A and the strong canonical signaling activity the two constructs exhibited with TOPFlash assay (Fig. 2b). In the case of FZD4_3linker and FZD4_6linker, which

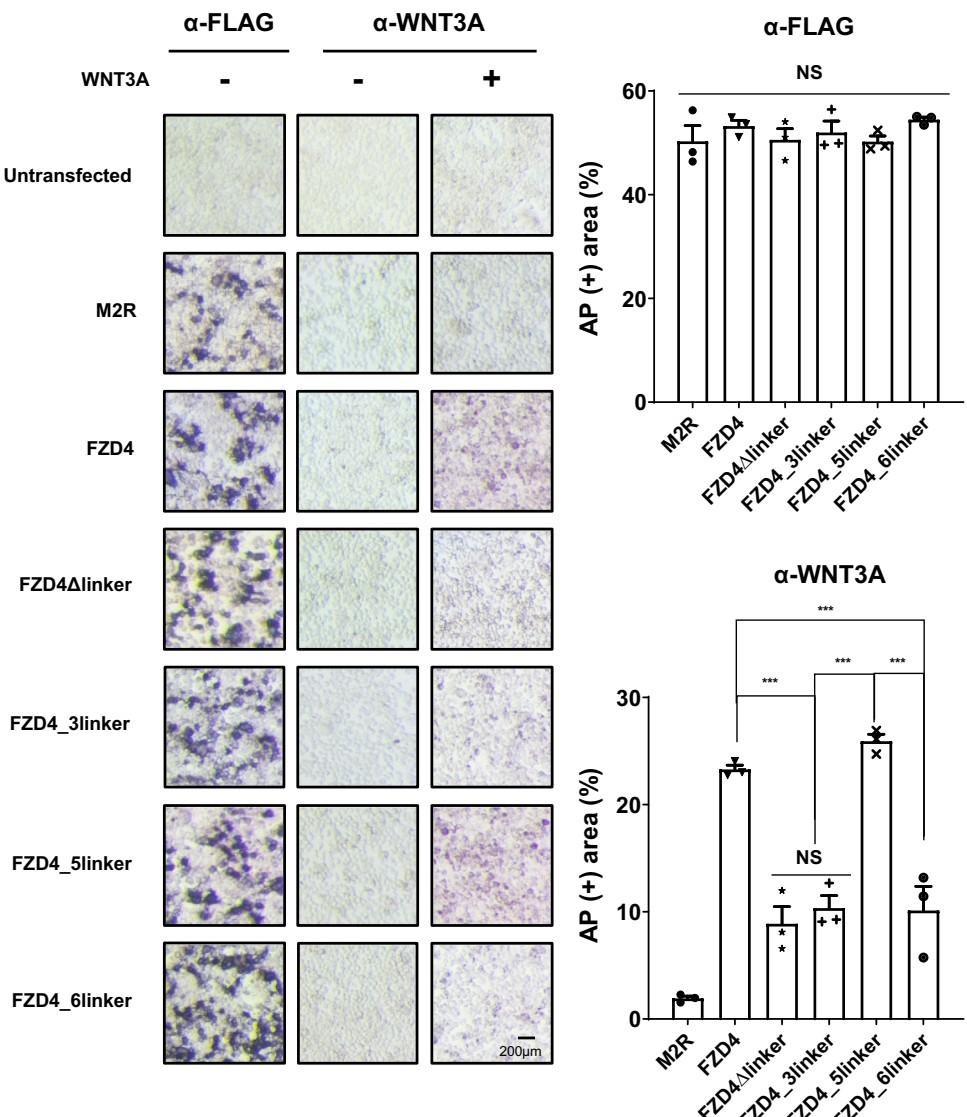

**Fig. 3 FZD linker domain regulates WNT3A binding to FZD on cell surface.** AP-stained images of dFZD1-10$^{-/-}$ cells transfected with target receptors, with or without L3a treatment. Surface-expressed receptors and receptor-bound WNT3A were detected by AP-conjugated anti-FLAG and anti-WNT3A antibodies, respectively. Bound antibodies were subsequently visualized using NBT/BCIP substrates (Scale bar: 200 μm). The AP-stained area was quantitatively analyzed using ImageJ and displayed as a bar graph. The error bar indicates the SD (standard deviation) of three replicates. Statistical comparisons were performed using one-way ANOVA followed by the Tukey's test. 'NS' represents 'not significant' and '***' represents $P < 0.001$.

demonstrated dramatically decreased responses to WNT3A, there was little change in the oligomerization of FZD upon WNT3A treatment (Fig. 4a).

Our results show that while the CRD is the main ligand-binding site, our immunostaining experiments with FZDΔlinker, FZD4_3linker, and FZD4_6linker indicate that for Wnt, like Norrin, the linker plays a role in ligand recognition. However, the difference in binding was not as drastic as that observed for signaling activity. One reason for this could be that the AP-staining protocol is limited in sensitivity, as WNT3A binding is detected through indirect staining of FZD-bound WNT3A. Another possibility is that the effect of the linker on signaling may not be solely due to a change in ligand-binding. This has been discussed in more detail below.

**LRP6 and DVL2 recruitment is affected by linker-swapped mutations.** For canonical Wnt signaling LRP5/6 is an essential co-receptor, the recruitment of which has been reported to be

sufficient to activate signaling[35,36]. To monitor the interaction between FZD mutants and LRP6 at the basal level, as well as upon Wnt activation, BRET assays were performed in both the presence and absence of the Wnt ligand. Again, a titration assay was performed to determine the optimal ratio of BRET acceptor/donor (Supplementary Fig. 8). FZD6 was included as a negative control and it showed a low level of constitutive colocalization with LRP6, reflecting its exclusively noncanonical signal activating tendency, although the BRET ratio was higher than that of muscarinic receptor (M2R) (Supplementary Fig. 9). With WNT3A, LRP6 was recruited to FZD4 and the FZD4_5linker mutant. Other linker-swapped and linker deletion mutants of FZD4 showed reductions in LRP6 recruitment (Fig. 4b). These results demonstrate that the reduced signaling activity of FZD4_6linker is the combinatorial effect of disruption in binding with WNT3A and a reduced association with the co-receptor, LRP6.

We also performed a BRET assay, preceded by titration assay for the BRET pair (Supplementary Fig. 10), to monitor the

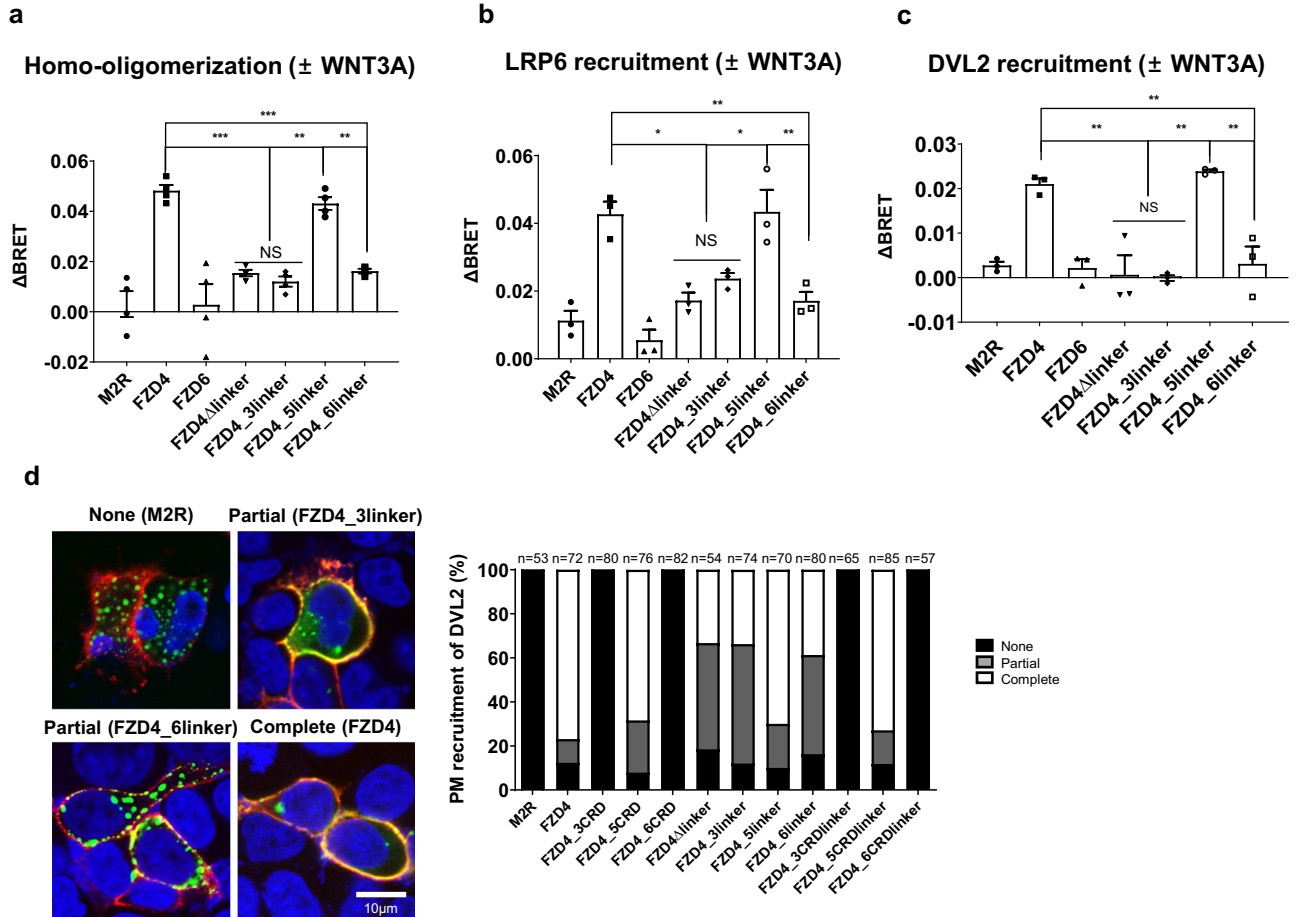

**Fig. 4 Reduction of FZD4 oligomerization and LRP6 and DVL2 recruitment by FZD linker mutation.** BRET assays were performed with (**a**) FZD-Rluc/FZD-eYFP, (**b**) FZD-Rluc/LRP6-eYFP, and (**c**) FZD-Rluc/DVL2-eYFP pairs to monitor homo-oligomerization, LRP6 recruitment, and DVL2 recruitment, respectively, in response to WNT3A. A functionally unrelated muscarinic receptor 2 (M2R) was used as a negative control. The titration curve for each pair is demonstrated in Supplementary Figs. 6, 8, and 10, respectively. Bar graphs are drawn using the ΔBRET value for each construct, calculated by subtracting the BRET ratio of afamin-treated sample from that of WNT3A-treated sample. The error bars of all bar graphs are the SEM of three independent experiments. Statistical comparisons were performed using the two tailed *t*-test. 'NS' represents 'not significant', '***' represents *P* < 0.001, '**' represents *P* < 0.01, and '*' represents *P* < 0.05 for the mediated BRET assay on various FZD linker-swapped mutants. **d** IF images of cells transfected with M2R, FZD4 wild-type, and FZD4 linker mutants are shown as examples of 'none', 'partial', and 'complete' recruitment of DVL2 (Scale bar: 10 μm). The graph shows the percentage of differential DVL2 recruitment state to FZD. The number of counted cells per condition is indicated.

recruitment of DVL2 to FZD in response to WNT3A. At the basal state, there was no significant difference in DVL2 recruitment level among different FZD constructs (FZD3, FZD4, FZD6, FZD4_3linker, FZD4_5linker, and FZD4_6linker), all of which were higher than M2R (Supplementary Fig. 11). However, upon WNT3A treatment, only FZD4 and FZD4_5linker, which were the two constructs that displayed TOPFlash activity, successfully recruited DVL2 (Fig. 4c). The recruitment was further confirmed through IFA, which revealed localization of cytosolic DVL2 at the membrane (Supplementary Figs. 12, 13). We classified the recruitment to three categories, 'none', 'partial', and 'complete' and plotted bar graphs pertaining to the observed DVL2 recruitment state for each construct in response to WNT3A (Fig. 4d).

Altogether, the FZD4 linker appears to be important for WNT3A binding, FZD oligomerization, LRP6 recruitment, and DVL2 recruitment, which have a combinatorial effect on canonical signal activation.

**FZD6 mutant containing the FZD4 CRD/linker responds weakly to the canonical ligand**. As we observed that FZD4 interacts and responds selectively to WNT1 and WNT3A through a

combination of its CRD and linker, we were curious whether chimeric mutants of the noncanonical receptors, FZD3 and FZD6 (FZD3/6) containing the canonical FZD4 CRD or CRDlinker, could activate canonical signaling. Thus, we designed chimeric mutants of FZD3/6, in which the CRD or CRDlinker were replaced by the corresponding region of FZD4 (Fig. 1a and Supplementary Table 1). The surface expression and expression levels of each mutant were confirmed by IFA, flow cytometric analysis, and surface ELISA (Supplementary Figs. 2–4). We tested the canonical signaling activity of these constructs using all three canonical ligands: WNT1, WNT3A, and Norrin. The canonical signaling activity of these chimeric mutants was negligibly low compared to that of FZD4 (~10-fold). However, upon closer inspection, we observed that FZD3_4CRDlinker and FZD6_4CRDlinker (FZD3/6_4CRDlinker) consistently showed a 3 to 4-fold increase in activity for all three ligands (Fig. 5a, Supplementary Fig. 14). In particular, when recombinant WNT3A protein was treated, the difference in signaling activity between FZD3/6_4CRDlinker and FZD4 became less drastic (~3-fold), as the exposure to WNT3A was more controlled (Fig. 5b). It should be noted that the FZD6_4CRD mutant failed to elicit any response, again highlighting the importance of both CRD and the linker in signaling.

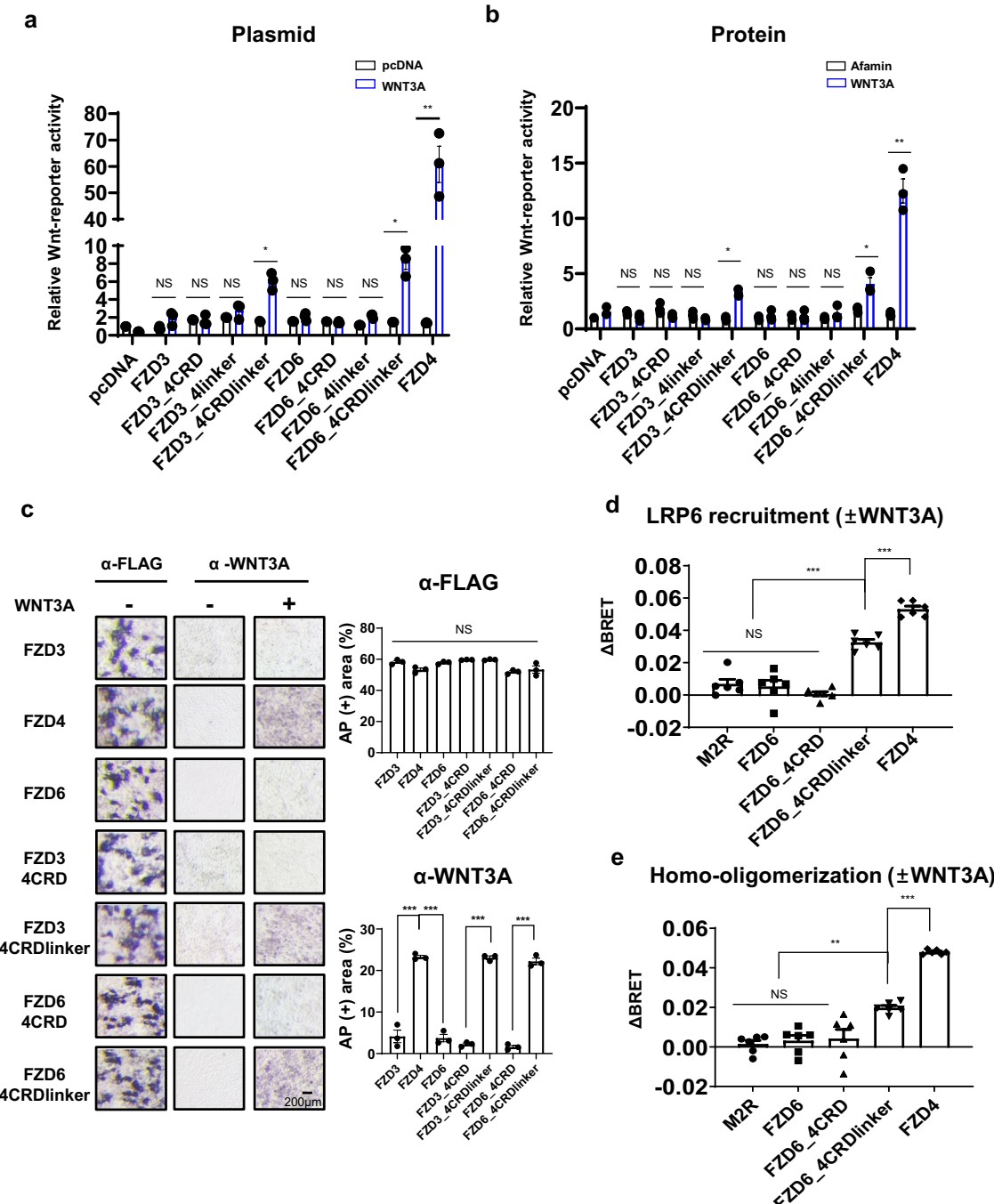

**Fig. 5 Implantation of canonical FZD4 ligand-binding site in noncanonical FZD3/6.** The effects of substitution of the FZD4 CRD, linker or CRDlinker into the noncanonical FZD subtype were tested with (**a**) WNT1 and WNT3A plasmid and (**b**) recombinant WNT3A protein. TOPFlash assay results are plotted as bar graphs with error bars indicating the SEM of three independent experiments. **c** AP staining was performed with an AP-conjugated anti-FLAG and anti-WNT3A antibodies to observe the surface expression of FZD and WNT3A bound to cells expressing indicated FZD (Scale bar: 200 μm). The AP-stained area was measured using ImageJ and displayed as a bar graph with error bars indicating the SD of three replicates. The ΔBRET ratio was measured and calculated for (**d**) FZD-Rluc/LRP6-eYFP pair to monitor FZD-LRP6 interaction and (**e**) FZD-Rluc/FZD-eYFP pair to monitor homo-oligomerization of FZD in response to WNT3A. Error bars represent the SEM of six independent experiments. Statistical comparisons were performed using one-way ANOVA followed by the Tukey's test. 'NS' represents 'not significant', '***' represents $P < 0.001$, '**' represents $P < 0.01$, and '*' represents $P < 0.05$.

We investigated the binding of WNT3A to FZD3/6_4CRD and FZD3/6_4CRDlinker using AP staining assays. As expected, in contrast to FZD3/6_4CRD, which failed to bind to WNT3A, FZD3/6_4CRDlinker bound to WNT3A (Fig. 5c). We did not observe a substantial difference in AP stain intensity between FZD3/6_4CRDlinker and FZD4, but the canonical signaling strength of FZD3/6_4CRDlinker was insignificant compared to

that of FZD4, perhaps due to the failure to properly recruit LRP6 (Fig. 5d). Of note, in contrast to FZD4, FZD6 and its variants showed no recruitment of LRP6 at the basal state (Supplementary Fig. 15). In addition, BRET signal for homo-oligomerization of FZD6_4CRDlinker upon WNT3A treatment was much lower than that for FZD4 homo-oligomerization (Fig. 5e, Supplementary Fig. 15).

Therefore, our results suggest that the swapping of the CRD and linker may, to a certain extent, direct FZD towards a signal response that is not naturally inherent in that FZD subtype by conferring ligand-binding ability, but other parts besides the extracellular domain may also be required for efficient signal transduction into the cytoplasm.

**Effect of FZD linker on noncanonical Wnt signaling.** So far, we have shown that the linker domain is involved in canonical Wnt signaling via WNT1 and WNT3A, and that it does so by affecting the ligand-binding and LRP6 recruitment. We were curious if the linker had the same impact on the noncanonical pathway. While it is well known that canonical Wnt signaling activity can be measured by the TOPFlash assay, the cell-based assay system for noncanonical Wnt signaling has not yet been well established, in part due to the diversity in signaling. It has been previously shown that for WNT5A-mediated noncanonical signaling, the ERK, JNK, and AKT pathways can be affected, resulting in different phosphorylation levels of the three proteins, which can be investigated by western blot analysis[37,38]. FZD1 subtype linker was chosen to generate chimeric mutants of FZD3/6 as FZD1 is a canonical FZD reported to have no interaction with WNT5A[39]. Similar to the process of construction of the FZD4 chimeric mutants, the CRD or linker-swapped chimeric mutants of FZD3/6 containing FZD1 CRD or linker was designed to test the effect of the linker on noncanonical signaling (Supplementary Table 1).

We analyzed the phosphorylation levels of ERK, JNK, and AKT by western blotting after transfection with FZD3/6 chimeric constructs. Both FZD3/6_1linker mutants increased the phosphorylation levels of ERK, JNK, and AKT to varying degrees in response to WNT5A, but at lower levels of phosphorylation compared to their wild-type counterparts (Fig. 6a, Supplementary Fig. 16). To determine if FZD3/6 and FZD3/6_1linker varied in WNT5A binding, AP staining assays were performed. Although FZD3/6_1linker mutants showed a certain degree of AP staining, they displayed less intense staining than those of FZD3/6 (Fig. 6b, Supplementary Fig. 17). Of note, FZD3/6_1CRD did not show binding to WNT5A, similar to FZD1. This suggested that the FZD6 CRD plays a key role in the specific interaction with WNT5A, but the FZD6 linker also contributes to WNT5A binding. As the effect of the linker on WNT5A binding to the noncanonical receptor became apparent, we also performed BRET assays on FZD6_1linker to measure the degree of DVL2 recruitment and homo-oligomerization of FZD induced by WNT5A (Supplementary Fig. 18). Compared to FZD6, FZD6_1linker showed lower levels of WNT5A-induced homo-oligomerization (Fig. 6c, Supplementary Fig. 19), and DVL2 recruitment was also negatively affected by the linker swap (Fig. 6d, Supplementary Fig. 20). As before, the colocalization of DVL2 in response to WNT5A was further confirmed by IFA analysis (Supplementary Figs. 21, 22). While DVL2 was colocalized with FZD6, FZD6_1linker showed increased count of partially recruited DVL2 compared to that of FZD6 (Fig. 6e).

Altogether, the linker domain affects FZD homo-oligomerization and DVL2 recruitment of the noncanonical FZD subtypes, FZD3 and FZD6, in the presence of WNT5A. Since we showed only the changes in the phosphorylation levels of ERK, JNK, and AKT upon WNT5A treatment, a more systematic analysis is required to demonstrate the essentiality of the linker domain for noncanonical Wnt signaling in general.

## Discussion

In summary, we demonstrated that the linker domain of FZD contributes to Wnt-FZD interaction and Wnt signal transduction to the cytoplasm. TOPFlash assays using various FZD4 linker mutants showed decreased luciferase activity in FZD4_3linker and FZD4_6linker compared with FZD4_5linker and FZD4. Our cell-based binding assays revealed that the weaker signaling capacity in FZD4_3linker and FZD4_6linker was linked to decreased binding to WNT3A, suggesting that the linker domain is involved in the interaction with the Wnt ligand. Perhaps due to less binding of ligand, FZD4_3linker and FZD4_6linker exhibited reduced capacity to homo-oligomerize and recruit co-receptor LRP6. Similar results of signal attenuation in the noncanonical Wnt signaling pathway were obtained when FZD6_1linker was stimulated with a noncanonical WNT5A ligand. Our results demonstrate that the linker plays a general role in ligand recognition and downstream signaling. A recent publication of cryo-EM studies of FZD5/XWnt8 with fiducial markers failed to resolve the density of ligand-bound extracellular regions, and thus far, structural data on how the Wnt ligand interacts with full-length FZD are not yet available. FZD/Wnt interactions appear to be very dynamic in terms of orientation between the CRD/Wnt region and transmembrane domain, as the linker allows considerable movement. As the FZD linker is highly diverse in both sequence and length and given the promiscuity of Wnt/FZD interactions, we suspect that the interaction is not sequence-specific but is based on general features of sequence composition such as electrostatic charges or hydrophobicity; however, this needs further investigation.

Wnt stimulation of FZD3_4CRDlinker and FZD6_4CRDlinker showed that noncanonical FZD can be steered to activate canonical signaling, although the activity is much lower than that of the wild-type FZD4. Notably, even Norrin, an FZD4 specific ligand, could elicit responses from FZD5_4CRDlinker and FZD6_4CRDlinker. Simple swapping of the CRDlinker domain, but not the CRD alone, resulting in switching of the activated downstream signal from its native signaling pathway, may seem surprising at first, but considering that CRDlinker is the major ligand-binding site, it is certainly conceivable. Indeed, FZD6_4CRDlinker had no problem binding to WNT3A, regardless of the transmembrane region, as shown by AP staining. If the extracellular ligand/CRD complex was sufficient to recruit LRP5/6 and fully activate canonical signaling, the Fzd6_4CRDlinker mutant had the capacity to do so. However, our results suggest that there may be other key elements of signal activation in the transmembrane and intracellular regions required for the full activation of canonical signaling. Recently, Kozielewicz et al. reported a study where they used a NanoBiT/BRET system to assess the importance of the core FZD transmembrane domain in interaction with WNT3A[40]. Their study yielded mixed results, implying that the transmembrane core could contribute positively or negatively to WNT3A binding, depending on the FZD subtype. In any case, the results converged on the involvement of the core domain in ligand interaction. However, it is noteworthy that the CRDlinker chimeras used in their study are different from ours in that their CRDlinker domain was designed to contain all three conserved Cys residues in the linker domain. FZD/Wnt interaction appears to be more complex and dynamic than the previously assumed model of CRD as an independent ligand-binding site. The molecular basis of this interaction and how it varies among FZD and Wnt subtypes is critical in understanding the activity and selectivity of Wnt signaling and requires further evaluation.

Our findings on the FZD6 linker mutants showed that the effect of the linker is to some extent present even in noncanonical signaling, as visualized through the change in the efficiency of downstream protein phosphorylation, DVL2 recruitment, and FZD oligomerization. The FZD6_1linker showed reduced signaling capacity compared to FZD6. The importance of the FZD6 linker has been noted previously, as it has been shown that CRD-deleted FZD6, as long as conserved 'triad' of cysteines in the linker are

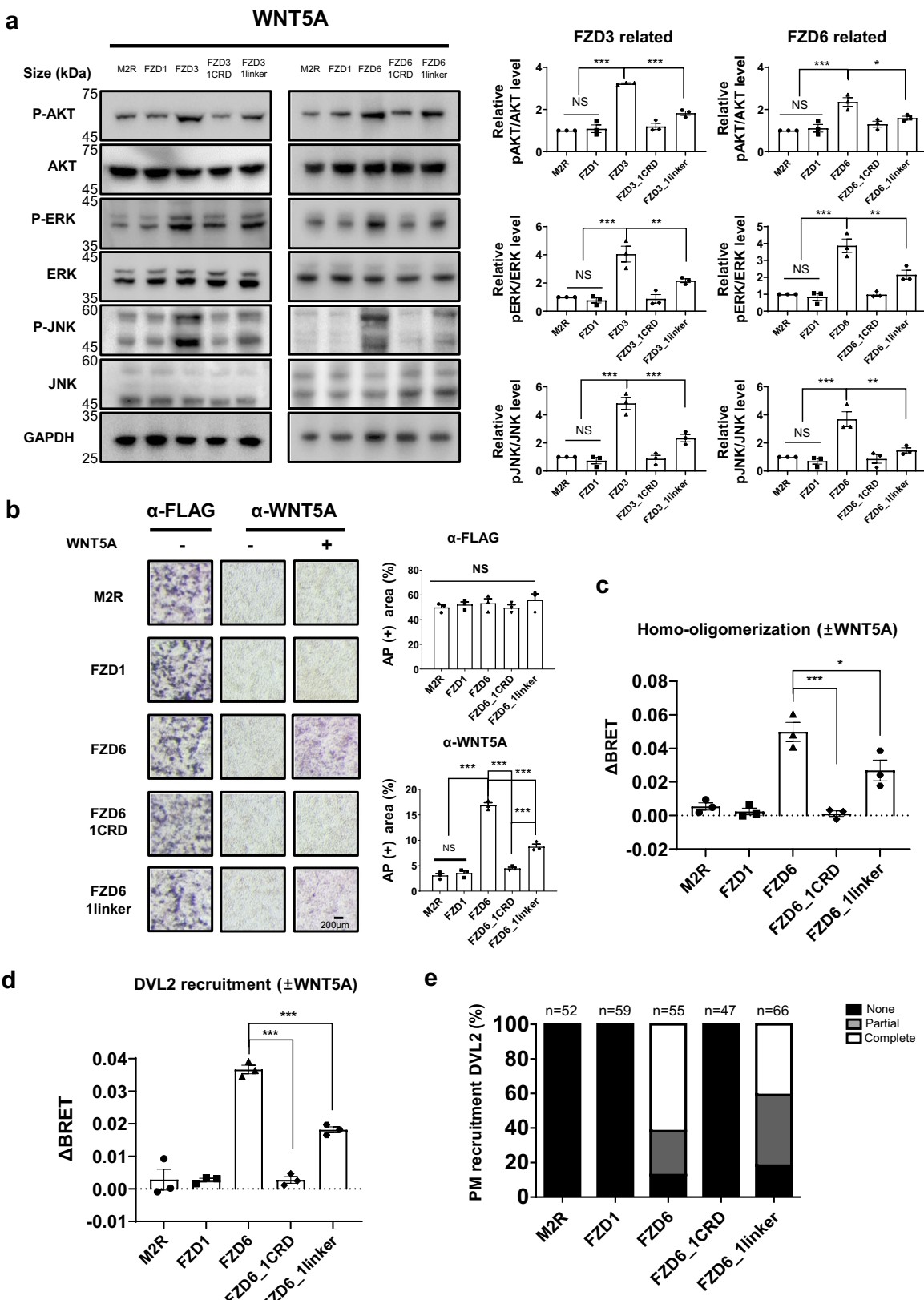

intact, retains the ability to induce the electrophoretic mobility shift of DVL2, which is an indication for activated DVL2[22]. Our data supports this in part because the FZD6_1linker, which still contains the conserved cysteines, successfully recruited DVL2, although to a lesser extent than the FZD6 wild-type. However, our results differ in a way, as we believe that the CRD is crucial in recognizing Wnt and

recruiting DVL2. DVL2 recruitment was clearly shown to be a WNT5A-mediated response, and our noncanonical signaling study demonstrated that CRD is indispensable for the Wnt-mediated reaction. The reasons for these differences need to be further studied. However, both studies agree that the linker domain is important for signaling.

**Fig. 6 FZD linker domain affects noncanonical pathway.** Effects of the FZD linker domain on the noncanonical pathway. **a** Western blot analysis of the phosphorylation levels of ERK, AKT, and JNK in response to WNT5A in dFZD1-10$^{-/-}$ cells expressing the indicated FZD mutants. The error bars indicate SEM of measurements from three independent western blots (Supplementary Fig. 16). The phosphorylation level of each protein was quantified using ImageJ, and the phosphorylated protein levels relative to the non-phosphorylated protein levels are shown as a bar graph. Statistical comparisons were performed using one-way ANOVA followed by the Tukey's test. 'NS' represents not significant, '***' represents $P < 0.001$. **b** AP-stained images of transfected dFZD1-10$^{-/-}$ cells with the FZD6 CRD and linker-swapped mutants, with or without L5a treatment (Scale bar: 200 µm). Bar graphs represent AP-stained surface area quantified using ImageJ. The error bars indicate the SD of three replicates. Anti-FLAG stained area represents surface-expressed FLAG-tagged FZDs, while anti-WNT5A stained area would represent WNT5A bound receptors. Statistical comparisons were performed using one-way ANOVA followed by the Tukey's test. 'NS' represents 'not significant' and '***' represents $P < 0.001$. The BRET assay was performed to monitor (**c**) FZD homo-oligomerization and (**d**) DVL2 recruitment upon WNT5A treatment with FZD-Rluc/FZD-eYFP and FZD-Rluc/DVL2-eYFP pairs, respectively. The titration curve for each pair is demonstrated in Supplementary Fig. 18. The ΔBRET ratio was plotted with error bars indicating the SEM of three independent experiments. Statistical comparisons were performed using the two-tailed $t$-test. 'NS' represents 'not significant', '*' represents $P < 0.05$, and '***' represents $P < 0.001$. **e** Recruitment of DVL2 was classified as 'none', 'partial', and 'complete' (Supplementary Figs. 21, 22). The graph shows the percentage of differential DVL2 recruitment state to FZD. The number of counted cells per condition is indicated.

---

The findings of this research establish the linker as a functional domain that contributes to Wnt binding and signaling. These results suggest that the linker could be a modulation site for Wnt signaling. This has great potential as the FZD linker is unique in each subtype and provides the possibility of having FZD subtype-specific modulators capable of fine-tuning the Wnt signal. Wnt signaling is a complex system that has a myriad of effects throughout the body and has been linked to various diseases, including cancer; however, its intertangled signaling pathways have made it extremely challenging to target this pathway for drug development. The prospective of having a specific target site for each FZD subtype would greatly aid in the design of novel and safe drugs to treat diseases caused by aberrant Wnt signaling.

## Methods

**Cell culture.** The FZD null cell line (dFZD1-10$^{-/-}$ HEK293T cells) were kindly gifted by Dr. Vanhollebeke from the Université libre de Bruxelles (ULB), Belgium. Cells were cultured in Dulbecco's modified Eagle's medium (DMEM) (Biowest, France) supplemented with 10% FBS (Biowest, France) and 1% antibiotic/anti-mycotic (Thermo Fisher Scientific, USA) in a humidified 5% CO$_2$ incubator at 37 °C.

**Construct design.** All FZD constructs were subcloned into a pcDNA3.1 vector with an N-terminal preprotrypsin leader, followed by a FLAG tag. All FZD mutant constructs used in this study are summarized in Supplementary Table 1. For the linker swap constructs, the FZD4 linker region (161–203) was swapped with the corresponding regions of FZD3 (136–188), FZD5 (150–221), and FZD6 (132–184), respectively. For the CRD swap constructs, the FZD4 CRD region (41–160) was swapped with the CRDs of FZD3 (23–135), FZD5 (27–149), and FZD6 (19–131), respectively. For the CRDlinker swap constructs, the FZD4 CRDlinker region (41–203) was swapped with the corresponding regions of FZD3 (23–188), FZD5 (27–221), and FZD6 (19–184), respectively. Chimeric FZD constructs were made by overlap PCR and digestion with an appropriate restriction enzyme, as follows. Not1-HF and BamHI were used for FZD1, Not1-HF and XhoI for FZD3/4/5, and Not1-HF and EcoRI for FZD6. The digested insert DNAs were ligated into pcDNA3.1 vector, also digested with the corresponding enzymes. For BRET assays, Rluc8 and eYFP were fused to the C-terminus with a GGSGG linker, respectively, and DVL2 was fused with eYFP to the N-terminus with a GGSGG linker. All PCR primers used for cloning are listed in Supplementary Data 2 file.

**Immunofluorescence assay.** dFZD1-10$^{-/-}$ HEK293T cells were seeded in a 35 mm glass bottom confocal dish and transfected with the indicated FZD construct using Metafectene Pro (Biontex, Germany). Forty-eight hour after transfection, the cells were fixed with 4% paraformaldehyde (Sigma, USA) at room temperature for 10 min. Fixed samples were washed twice with PBS and blocked with 5% bovine serum albumin (BSA) for 30 min. As all FZD constructs have a FLAG tag at their N-terminus, cell surface localization was detected with an anti-FLAG antibody (Cell Signaling Technology, USA). The primary antibody was incubated overnight at 4 °C. The next day, cells were washed three times with PBS and stained with a secondary antibody conjugated to Alexa555 and Hoechst 33343. For the DVL2 recruitment assay, dFZD1-10$^{-/-}$ HEK293T cells were co-transfected with eYFP-DVL2 and each indicated FZD construct. After 48 h transfection, the transfected cells were incubated with either LSL or L3a conditioned media for 10 min at 37 °C. All samples were visualized using an LSM 700 confocal microscope (Carl Zeiss, Germany), and images were processed using Zen lite software (Carl Zeiss, Germany).

**BRET assay.** dFZD1-10$^{-/-}$ HEK293T cells were co-transfected with Rluc8 and eYFP constructs at a 1:5 ratio. Forty-eight hour post-transfection, cells were detached with PBS supplemented with 20 mM EDTA and distributed to a white 96-well microplate. Each experiment was performed with at least six replicates. Substrate, coelenterazine h (NanoLight Technology, USA), was added to each well to a final concentration of 5 µM. For the ligand-induced response, where appropriate, recombinant WNT3A (in complex with afamin) and WNT5A (R&D Systems, USA) were added to final concentrations of 400 ng ml$^{-1}$ and 500 ng ml$^{-1}$, respectively, using an automated injector. The BRET signal was determined as the ratio of light emitted by the energy acceptor (YFP, 535 nm) to light emitted by the energy donor (Rluc8, 488 nm). Ligand-induced ΔBRET was calculated by subtracting the BRET ratio of the vehicle-treated sample from that of the ligand-treated sample for each pair. All BRET signals were collected using a Mithras LB940 instrument (Berthold Technologies, Germany), and graphs were plotted with GraphPad Prism 8.

**TOPFlash luciferase assay.** dFZD1-10$^{-/-}$ HEK293T cells were plated in a white 96-well dish, and the total amount of transfected DNA was matched to 200 ng per well. Metafectene Pro (Biontex, Germany) was used as the transfection reagent, and each experiment was performed with at least three replicates. Co-receptor LRP6, TCF/LEF luciferase reporter for canonical signal, and Renilla luciferase as a control to normalize the luminescence signal of TCF/LEF luciferase were co-transfected with the indicated FZD construct with or without Wnt ligand. At 24 h post-transfection, the media was changed to growth media to enhance protein expression, and the luciferase signal was measured 48 h after transfection. The ligand WNT1 or WNT3A was co-transfected together with the receptor or in the case of WNT3A, where indicated, the ligand was treated exogenously as a recombinant protein. WNT3A protein as a complex with afamin or afamin alone as a control was added during the media change step to a final concentration of 400 ng ml$^{-1}$. Luminescence was measured using a microplate luminometer Mithras LB940 instrument (Berthold Technologies, Germany). The normalized signal was determined by dividing the luminescence signal intensity of TCF/LEF luciferase by that of Renilla luciferase, and the relative signal was determined by setting the pcDNA-transfected vehicle-treated group as the reference point of one. Graphs were plotted using GraphPad Prism 8.

**Alkaline phosphatase (AP) staining assay.** The protocol for the AP staining assay was described in detail by Cho et al.[30,31]. Briefly, dFZD1-10$^{-/-}$ HEK293T cells were seeded on Poly-D-Lysine-coated dishes. At 48 h after transfection with the indicated mutant constructs, the transfected cells were either stained with anti-FLAG antibody (Cell Signaling Technology, USA) or incubated with indicated ligand for 2 h at 4 °C. The ligand-treated cells were then stained with anti-WNT3A antibody (Thermo Fisher Scientific, USA) for 2 h at 4 °C. The cells were then washed 5–6 times with cold PBS, fixed with 4% PFA, and heat denatured at 70 °C for 1 h. The cells were stained with diluted AP conjugated secondary antibody, and cell surface-bound AP was visualized by incubation with an NBT/BCIP substrate (Thermo Fisher Scientific, USA) at room temperature. AP-stained areas were measured through digital photographs using an ImageJ (NIH) with at least three images from each dish. Briefly, the images were first converted to 8-bit greyscale and then to binary images with a threshold value to define positively stained areas against unstained areas. An identical threshold value was set for all the images. The positively stained areas were assessed using the 'Analyze particles' function. The percentage of stained area was plotted with GraphPad Prism 8.

**Surface ELISA assay.** dFZD1-10$^{-/-}$ cells were incubated at 37 °C for 48 h after transfection with expression plasmid containing each FZD mutant. Whole-cell ELISA assays were performed as described before[41] using clear 96-well plates (SPL, Republic of Korea). Each well was treated with 4% paraformaldehyde (PFA, T&I, Japan) for fixation and washed with PBS. The cells were incubated with blocking

solution containing 5% bovine serum albumin (BSA, Bovogen, Australia) for 30 min. After removal blocking solution, anti-FLAG antibody (Cell Signaling Technology, USA) was treated to each well. Subsequently, the cells were treated with HRP-conjugated secondary antibody (Enzo, USA) and incubated for 2 h, after which TMB solution (Thermo Fisher Scientific, USA) was added to each well and incubated until light-blue color was observed. The stain-containing supernatant was moved to another plate, and further reaction was blocked by adding 2 M HCl. The absorbance at 450 nm was measured using a FlexStation 3 system (Molecular Devices, USA). For normalization, the remaining cells in the plate were additionally treated with Janus Green solution (0.2% w/v, TCI, Japan) followed by a PBS wash. Any excess stain was washed out with distilled water and quenched with 0.5 M HCl. Absorbance was read at 595 nm using the FlexStation 3 system (Molecular Devices, USA). The normalized expression level of the receptor at the cell surface was calculated by the ratio of the absorbance at 450 nm to that at 595 nm (A450/A595). The normalized signal was plotted as a bar graph using GraphPad Prism 8.

**Flow cytometry analysis.** For quantification of the surface-expressed receptor, each FZD mutant construct was transfected into dFZD1-10$^{-/-}$ cells. After transfection, the samples were detached using 20 mM EDTA in PBS and then resuspended with flow cytometry buffer (1% FBS + 0.1% BSA in PBS). The cells were incubated with PE-conjugated either anti-FLAG antibody or isotype control antibody (Cell Signaling Technology, USA) in flow cytometry buffer and incubated for 30 min at 4 °C. The stained cells were then washed and resuspended with the same buffer. Gate was set based on side scattering (SSC) and forward scattering (FSC) to exclude cell debris and dead cells. The cell count of the anti-FLAG antibody-stained sample and that of the isotype-stained sample were overlaid, and the area of the anti-FLAG stained sample that does not overlap with that of the isotype-stained sample was taken as the target protein expressing percentage of the cells. All events were acquired with FACS Canto (BD Biosciences, USA) and analyzed with the FACS FlowJo v10 software (BD Biosciences, USA).

**Purification of WNT3A/afamin and afamin.** Biologically active mouse WNT3A in complex with human afamin was prepared according to the method described previously[42]. Briefly, N-terminally PA-tagged WNT3A and N-terminally Target-tagged afamin were co-transfected into Expi293F cells to establish a stable cell line secreting the WNT3A/afamin complex, followed by the purification of the complex by using anti-PA tag NZ-1 antibody column[43]. Uncomplexed afamin was expressed in a similar manner and purified by anti-Target tag P20.1 antibody column[44]. The purified proteins were buffer-exchanged by dialysis against PBS (20 mM phosphate, 150 mM NaCl, pH 7.0) and stored at −80 °C until use.

**Western blot.** Cells transfected with each FZD mutant were incubated with L5a conditioned medium for 30 min before harvest and lysed with RIPA buffer containing protease inhibitors (protease inhibitor cocktail, Thermo Fisher Scientific, USA). The samples were heated at 95 °C for 1 min before SDS-PAGE gel loading. After protein transfer to PVDF, the membrane was washed with deionized water and incubated with blocking buffer (Smart-Block, Biomax, Republic of Korea) for 1 h. The blots were then incubated with each primary antibody, anti-JNK (1:1000, Cell Signaling Technology, USA), anti-pJNK (1:1000, Thermo Fisher Scientific, USA), anti-AKT (1:1000, Cell Signaling Technology, USA), anti-pAKT (1:1000, Cell Signaling Technology, USA), anti-ERK (1:1000, Cell Signaling Technology, USA), anti-pERK (1:1000, Cell Signaling Technology, USA), or anti-GAPDH (1:1000, Santa Cruz Biotechnology, USA) in TBST (TBS with 0.05% Tween-20) containing 2.5% BSA overnight at 4 °C. Each membrane was washed three times with TBST and incubated with HRP conjugated secondary antibody for 2 h at room temperature. The signals were visualized using West Glow$^{TM}$ FEMTO chemiluminescent substrate (Biomax, Republic of Korea) and detected with Chemidoc MP (Biorad, USA). Three independent repeats were done for each western blot. The band intensity was quantified using ImageJ (NIH). Graphs were plotted with relative ratio obtained by dividing the band intensity of each phosphorylated protein (pERK, pAKT, pJNK) by the band intensity of the corresponding total protein (ERK, AKT, JNK).

**Statistics and reproducibility.** At least three independent experiments were carried out for each assay. Statistical tests used for individual experiments are described in the corresponding figure legends. All statistical analyses were performed using one-way ANOVA followed by the Tukey's test, with $p < 0.05$ considered significant. Data analysis and figures were prepared with Excel (Microsoft) and GraphPad Prism 8.

**Reporting summary.** Further information on research design is available in the Nature Research Reporting Summary linked to this article.

**Data availability**

All data supporting the findings of this work are included in this article and the Supplementary Information file. Source data for all figures can be found in Supplementary Data 1 file and Supplementary Fig. 23. The newly generated plasmids

during this study were deposited to Addgene (FZD4_3CRD, 184252; FZD4_5CRD, 184253; FZD4_6CRD, 184254; FZD4-del-linker, 184255; FZD4_3linker, 184256; FZD4_5linker, 184257; FZD4_6linker, 184258; FZD4_3CRDlinker, 184259; FZD4_5CRDlinker, 184260; FZD4_6CRDlinker, 184261; FZD3_4CRD, 184262; FZD3_4CRDlinker, 184263; FZD5_4CRDlinker, 184264; FZD6_4CRD, 184265; FZD6_4CRDlinker, 184266). A list of PCR primers used for cloning is provided as a Supplementary Data 2 file. All other data are available from the corresponding author on reasonable request.

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

## Acknowledgements

This work was supported by the National Research Foundation of Korea (grant number NRF-2020R1A2C2003783) and Bio & Medical Technology Development Program (grant number NRF-2019M3E5D6063903) funded by the Korean government (to H.-J.C.). We thank Dr. Vanhollebeke from the Université libre de Bruxelles (ULB) for providing us with dFZD1-10⁻/⁻ HEK293T cells.

## Author contributions

S.-B.K. performed most BRET and cell-based assays with help from I.B. and Y.P., K.R., and C.K. performed flow cytometric analyses of FZD mutants. E.M. and J.T. purified the WNT3A ligand. I.B. and H.-J.C. conceived and directed the study. S.-B.K., I.B., and H.-J.C. outlined the manuscript and wrote the manuscript with contributions from all authors.

## Competing interests

The authors declare no competing interests.
