## [Peer Review File · Communications Biology]

Reviewers' comments:

Reviewer #1 (Remarks to the Author):

The manuscript by Choi and colleagues describe the role of the linker domain of Frizzled receptors and their effect on both canonical and non-canonical Wnt signalling. They select two canonical receptors (FZD4/5) and two non-canonical receptors (FZD3/6) and generate a number of chimeric mutants where they replace/delete the linker domain, the CRD domain, or both. They perform cell-based assays to measure Wnt-reporter activity, and use BRET for FZD homodimerization, and heterodimerisation with Dvl and LRP6 as well as utilizing flow cytometry, immunofluorescence, and western blotting. Overall, they demonstrate that the linker has a moderate and varying effect on the functional levels of canonical and non-canonical Wnt signalling, while these findings are interesting and important I believe some of the experiments need to be further elaborated and better analysed to be able to make solid conclusions from the data.

Specific points (Major):

1. Figure 1b – The authors provide relative surface expression levels of the various FZD constructs. What is the actual expression of FZD constructs (i.e are only 50% of the HEK cells expressing the constructs)? This data may be better represented by FACS histogram plots with respect to a non-transfected/no antibody control.
2. Figure 3 – the anti-FLAG antibody looks to have varying levels of expression across the different conditions – this could effect the overall interpretation of Wnt3a binding – is there a way to normalise this expression and obtain a quantification of the staining? There are numerous program available these days (e.g QuPath) which could be used for quantification of the AP staining. Furthermore, the authors should refrain from using and making any conclusions about binding affinity (even as a semi-quantitative measure) as no Kd can be measured from this assay.
3. Figure 5a&b) The data showing the Wnt reporter activation with 'canonical' CRDlinker swaps into 'non-canonical' FZD receptors are less than convincing. The actual increase in Wnt activity, both Wnt1 and Wnt3a (but in particular Wnt3a) is a very small fraction of FZD4 itself (as stated by the authors). The FZD3_4linker and FZD6_4linker are also both missing in the data. d) Flag antibody staining is also missing – again, in the text the authors should refrain from using 'binding affinity'.
4. Figure 6 – the western blots are of poor quality, with no quantification. Can a Fzd6 with Fzd1-CRD (FZD6_1CRD) also activate the non-canonical pathways (pAkt, pERK, pJNK) as it still has the FZD6 linker? Fzd3 is known to activate non-canonical signalling in response to Wnt5a (as stated by the authors) – Does a Fzd3 constructs with 'canonical' linkers fail to activate non-canonical signalling?
5. Supp Figure 2 – similar to Fig 1b – the data has been normalised to the FZD wildtype – although a better representation would be the raw expression levels of the constructs.
6. Supp Figure 5 – Control constructs of FZD4 with a 'non-canonical' linker and/or CRD combination should be investigated for Dvl recruitment by both IF and BRET if possible. Similar to Supp Figure 6 (below) lower magnification images with enough cells for analysis should be taken, presented (in combination with the high magnification images) and quantified.
7. Supp Figure 6 – the IF is not very convincing – few numbers of cells look like they are actually expressing the constructs – 1 cell has been picked out of the entire image – there is no quantification of the co-localisation and to my eye the colocalization of Fzd6_1linker +Wnt5a looks to be stronger than that of Fzd6 + Wnt5a itself. Lower magnification images with enough cells for analysis should be taken, presented (in combination with the high res) and quantified.

Specific Comments on Materials and Methods:

1. Generation of an FZD 1, 2, 4, 5, 6, 7, 8 null HEK293T cell line: Can the authors provide evidence of the generation of a FZD6-/- cell line by western blot, or FACS analysis?
2. Construct design – the authors should include primer sequences and method of cloning.
3. TOPflash luciferase assay – why was the media changed to growth media and Wnt1/Wnt3a were co-transfected – would it have no been better to serum starve and add recombinant Wnt1/Wnt3a – this would have been better to correct for transfection efficiency and ensure that the signal seen was primarily driven by the adding of Wnts – full growth serum can result in high levels of background TOPflash signal.

Specific Comments (Minor):

1. Citation of Wnt5a activation of FZD6 is a review (#20) – the authors should cite the original research article.
2. Supps Figure 3 – ANOVA and Tukeys were performed but there was no statistical analysis indicated on the graphs
3. Should include other studies (no in vitro and computational) on the study of the FZD linker domain. E.g (PMID: 33290484)
4. Manuscript should be thoroughly checked for typographical errors – e.g (2 µg ml⁻¹; Clontech) in “Generation of an FZD 1, 2, 4, 5, 6, 7, 8 null HEK293T cell line”

Reviewer #2 (Remarks to the Author):

Seung-bum Ko et al., present a study on the functional role of FZD linker in the WNT signalling. This work can be seen as a continuation of Bang et al., (2018) and Valnohova (2018) where the direct impact of different FZD-linker modifications was studied. The present work could be viewed as a substantial addition to the field if not for many methodological issues that in fact make the results questionable, and prevent the publication of the work in its current form. I will go Figure by Figure in my analysis:

Fig.1 While some colour changes can be done to make the Fig 1A more understandable to a reader, cell surface analysis should be re-done as HEK cells are notoriously bad for flow cytometric analysis (please refrain from using the term “FACS” as no cell sorting has been performed in the manuscript). Did the authors use median fluorescence intensity or mean fluorescence intensity? How did they assure that the cells were alive? Flow plots should be attached. Furthermore, if microscopic analysis of fluorescence was also attempted, then it is not quantitative, and prone to being subjective (authors show one transfected cell in each figure, which is not really convincing). The authors should use e.g. live-cell ELISA to get numbers representative of a whole-cell population in a well/on a dish. If the issue with the expression level of the receptors is not fixed, then it is impossible to understand the subsequent TOPFlash results as they are to some extent proportional to the cell surface expression levels of FZDs.

Fig. 2 The authors need to explain how the “Relative WNT-reporter activity” was calculated. In my opinion, since the authors are interested in WNT-induced effects, they should calculate the ratio of firefly to renilla signal for vehicle- and for WNT-treated and FZD-expressing cells. It is known that different FZDs produce different basal TOPFlash signal so it needs to be clearly stated what the graph represents. Additionally, please use dFZD1-10 HEK293T cells (Eubelen et al., 2018), they should be considered as a gold standard now in the field....What is the reason for creating/using a cell line that is not superior to what is already available?

Fig. 3 The AP analysis is a nice one but the WNT binding assessment can be substantially improved with NanoBRET. It will allow obtaining quantifiable data. Why didn't the authors use all the constructs they generated and ran AP on them?

Fig. 4 All the BRET data should be shown as titration curves with an increasing acceptor amount with or without WNTs. How do the authors know that the interactions are not bystander, e.g. that they do not appear to have a linear, non-saturable increase on a titration curve? BRET experiments are a beautiful way to study protein-protein interactions but they certainly require proper controls...

Fig. 5 Same issues as for Fig. 2 Please show anti-FLAG staining for Fig 5D.

Fig. 6A needs more comments in the text. How was the statistical analysis performed?

Immunoblots provide only semi-quantitative results....; Fig. 6B needs titration curves. If it happens to be true, then the authors should comment more on the apparent lack of FZD1-DVL proximity. I am not sure how the authors concluded that FZD-DVL belongs to the non-canonical pathway. Additionally, please state that DVL2 was used.

Supplementary figures: the issues are recurring; proper quantification of expression levels is crucial.

Other issues:

- Please use FZD not FZD receptors. FZDs are WNT-receptors.
- Please explicitly specify the number of technical replicates and independent experiments from which data are shown in the figures (for each figure).

The authors should also comment on:

- FZD3 and FZD6 WNT-induced TOPFlash signaling upon CRD+linker swap (Fig 5 A and B). Assuming the TOPFlash signal calculations are fine (or will be fixed and still shown the same trend), is it really a meaningful increase given the fact that FZD4 control increases the WNT signal about 60x (WNT-1) to ca. 120x (WNT-3A)? Anyhow, if FZD3 and FZD6 core can indeed signal via B-catenin dependent pathway upon ligand binding to a CRD-linker of another FZD, that will probably be the single most important finding of the paper, and one that will have an impact on the field.
- expression of their chimeric receptors (FZD4_6CRDlinker and FZD6_4CRDlinker) in comparison to the ones published by Koziellewicz et al., 2021. The reviewer realizes obviously that here different fusing points were used (several amino acids of the linker left with the receptor core in the present study).

My other comments:

- confocal images of DVL recruitment serve as a good addition to the BRET data.

Reviewer #3 (Remarks to the Author):

The authors have previously shown that the linker domain of Frizzled4 plays an important role in Norrin binding and signaling. This discovery was likely to lead them to investigate the role of the linker domain in the canonical and non-canonical Wnt signalings and in the molecular interactions that underlie them. Basically, their approach is unique and could provide new insights into the mechanisms underlying diversification of canonical and non-canonical Wnt signaling.

However, I have two major concerns with this paper. One is in their interpretation of BRET analysis. The other is about Frizzled that was used for comparison in the experiment shown in Figure 6.

First, I feel there is a problem with the interpretation of the BRET results shown in Figure 4a,b,c. I agree that there is a significant difference in delta BRET between the positive FZD4 and FZD4_5linkers and the negative ones containing FZD4_3linker and FZD4_6linker in the presence of Wnt3a. However, the delta BRET scores are also reduced in those positive for Wnt3a expression. Therefore, it is difficult to conclude from these data the molecular interactions between FZD4 and its partners as described by the authors. Strictly speaking, the oligomerization of FZD4 and FZD4_5linker and their interactions with LRP6 and Dvl are reduced by Wnt3a expression, but the reduction is smaller than that of FZD4_3linker and FZD4_6linker. If the authors want to claim that some specific linker domains actually affect ligand-induced FZD oligomerization, as well as ligand-induced FZD interaction with LRP6 and DVL, they should show evidence that these interactions are actually induced by ligands in their system.

Second, if we want to properly investigate the influence of linkers in the non-canonical Wnt pathway, it is highly desirable to compare the linker domains of FZD6 with those of FZD3, 4, and 6. I feel that this part is too preliminary and needs significant improvement if the authors want to include this part in the same paper.

In addition, their statement in Abstract, "Conversely, a FZD6 chimera, containing the FZD4 CRD and linker, binds to Wnt3a with similar affinity as FZD4 but does not produce a full canonical signal", may mislead readers. The point to be asserted here seems that "the FZD6 chimera, which contains the CRD and linker of FZD4, enhances Wnt canonical activity, ligand-dependent oligomerization, and interaction with LRP6 more than FZD6".

Specific points

Page 3, last sentence; The Wnt-Frizzled interaction has been studied by several groups. We recommend that you cite these papers as REFERENCE here. Wnt-Frizzled interaction has been investigated by several groups (Bhanot et al, 1996 ; Wu and Nusse, 2002; Takada et al, 2005).

Figure 1b, Supplementary Figure2b

It is unclear how the relative surface expression of each Fzd protein is measured: is the percentage of cells with Fzd protein on the surface relative to the entire transfected cell population, relative to the entire cell population expressing Fzd, or something else? What is the procedure for transfection and FACS analysis in these experiments?

Figure 2a,b; Authors should include data showing the Wnt activity of each Fzd-expressing cell in the absence of the Wnt ligand.

Page6, line13-14 "FzD4 and FZD5 responded to Wnt1 and Wnt3a, while FZD3 and FZD6 remained silent" : Since Figure 2b shows that FZD3 and FZD6 have higher Wnt activity than the negative control, pcDNA, the authors need to be careful in describing this result.

Page6, line15-16 "Replacing the FZD4 CRD with the CRD of FZD3 or FZD6 was shown to completely eliminate the activity of Wnt1 and Wnt3a"; Since Figure 2a shows that FZD4_3CRD and FZD4_6CRD have higher Wnt activity than the negative control, the phrase "completely eliminate" is not appropriate in this case.

Page 6, line 14; The data display for this statement is missing.

Page 7, line15; The detection of the interaction between Wnt and Frizzled by immunostaining was already reported in a paper that identified Frizzled as the Wnt receptor (Bhanot et al, 1996).

Figure 5d; FLAG staining is missing

Figure 5e,f; BASAL (without Wnt3a) is missing

Figure 6; Quantification results of Western blotting should be indicated.

Figure 6b; BASAL (without Wnt3a) is missing

We thank the reviewers for the suggestions and comments.

We greatly appreciate the time and efforts that the reviewers invested in reviewing our manuscript. Our revised manuscript has benefited from the insightful suggestions made by the reviewers. The major changes that we have made to our manuscript are given below.

- 1) We redid all cell-based experiments, such as TOPFlash and BRET assays, using “dFZD1-10^{-/-} HEK293T” cells, with the original author’s permission, as suggested by Reviewer #2.
- 2) For 26 FZD constructs, surface ELISA and flow cytometric analysis were performed to confirm the expression level of each construct (Supplementary Figures 3, 4).
- 3) Quantitative analyses of AP-stained images (anti-FLAG, anti-Wnt3a, anti-Wnt5a), Dvl2 recruitment, and western blots were performed (Figures 3, 4d, 5c, 6a, 6b, 6e, Supplementary Figures 5, 16).
- 4) Additional noncanonical signaling study using FZD3 chimeras was performed (Figure 6a, Supplementary Figure 16).

As detailed in our point-by-point responses given below (in blue font), we have addressed each of the points raised by the reviewers.

Reviewer #1 (Remarks to the Author):

The manuscript by Choi and colleagues describe the role of the linker domain of Frizzled receptors and their effect on both canonical and non-canonical Wnt signalling. They select two canonical receptors (FZD4/5) and two non-canonical receptors (FZD3/6) and generate a number of chimeric mutants where they replace/delete the linker domain, the CRD domain, or both. They perform cell-based assays to measure Wnt-reporter activity, and use BRET for FZD homodimerization, and heterodimerisation with Dvl and LRP6 as well as utilizing flow cytometry, immunofluorescence, and western blotting. Overall, they demonstrate that the linker has a moderate and varying effect on the functional levels of canonical and non-canonical Wnt signalling, while these findings are interesting and important I believe some of the experiments need to be further elaborated and better analysed to be able to make solid conclusions from the data.

Specific points (Major):

1. Figure 1b – The authors provide relative surface expression levels of the various FZD constructs. What is the actual expression of FZD constructs (i.e are only 50% of the HEK cells expressing the constructs)? This data may be better represented by FACS histogram plots with respect to a non-transfected/no antibody control.

We thank the reviewer for this suggestion. In the revised manuscript, flow cytometry histograms of FZD-expressing cells are included in Supplementary Figure 3. These histograms show the percentage of cells expressing the construct, and approximately 60–65% of the HEK cells expressed the FZD constructs.

2. Figure 3 – the anti-FLAG antibody looks to have varying levels of expression across the different conditions – this could effect the overall interpretation of Wnt3a binding – is there a way to normalise this expression and obtain a quantification of the staining? There are numerous program available these days (e.g QuPath) which could be used for quantification of the AP staining. Furthermore the authors should refrain from using and making any conclusions about binding affinity (even as a semi-quantitative measure) as no Kd can be measured from this assay.

We thank the reviewer for this suggestion. As suggested, the expression was quantified using the

ImageJ to estimate the area that was positively stained with alkaline phosphatase. The quantified AP (+) area for each construct was plotted as a bar graph and is shown in Figures 3, 5c, and 6b and Supplementary Figures 5 and 16.

We agree with the reviewer and have refrained from using the term 'binding affinity' in the revised manuscript.

3. Figure 5a&b) The data showing the Wnt reporter activation with 'canonical' CRDlinker swaps into 'non-canonical' FZD receptors are less than convincing. The actual increase in Wnt activity, both Wnt1 and Wnt3a (but in particular Wnt3a) is a very small fraction of FZD4 itself (as stated by the authors). The FZD3_4linker and FZD6_4linker are also both missing in the data. d) Flag antibody staining is also missing – again, in the text the authors should refrain from using 'binding affinity'.

We have added anti-Flag AP staining in Figure 5c and have included FZD3_4linker and FZD6_4linker in our Wnt-reporter assay, as shown in Figures 5a and 5b.

We agree with the reviewer's opinion that the signals caused by FZD3/6_4CRDlinker are rather small compared to that of FZD4. However we would like to note that the signals were observed to be at a statistically significant level, and given that FZD3/6 are strictly noncanonical Wnt receptors, this is no small feat, especially when compared to the absence of a signal from FZD3/6_4CRD. In addition, during revision we discovered that when we treated the cells with purified recombinant Wnt3a instead of co-transfection with the Wnt3a DNA, signaling difference between FZD3/6_4CRDlinker and FZD4 became less drastic, as demonstrated in Figures 5a and 5b. We believe that this difference is caused by a longer and more excessive exposure to Wnt3a when DNA was co-transfected. Together with our AP-staining data showing Wnt3a binding to FZD3/6_4CRDlinker, these data suggest that there may be other key factors besides ligand binding to the extracellular region of FZD to efficiently transmit the signal into the cytoplasm. While we do not have a clear explanation at this point, we do believe that this is an interesting piece of data in understanding the molecular basis of Wnt signaling activation.

We discussed about this issue in p10-11 as below.

p10 : *"In particular, when recombinant Wnt3a protein was treated, the difference in signaling activity between FZD3/6_4CRDlinker and FZD4 became less drastic (~3-fold), as the exposure to Wnt3a was more controlled (Fig. 5b). It should be noted that the FZD6_4CRD mutant failed to elicit any response, again highlighting the importance of both CRD and the linker in signaling."*

p11: *"Therefore, our results suggest that the swapping of the CRD and linker may, to a certain extent, direct FZD towards a signal response that is not naturally inherent in that FZD subtype by conferring ligand-binding ability, but other parts besides the extracellular domain may also be required for efficient signal transduction into the cytoplasm."*

4. Figure 6 – the western blots are of poor quality, with no quantification. Can a Fzd6 with Fzd1-CRD (FZD6_1CRD) also activate the non-canonical pathways (pAkt, pERK, pJNK) as it still has the FZD6 linker? Fzd3 is known to activate non-canonical signalling in response to Wnt5a (as stated by the authors) – Does a Fzd3 constructs with 'canonical' linkers fail to activate non-canonical signalling?

We improved the quality of the western blots, and the phosphorylation level of each protein was quantified using the ImageJ. In Figure 6a, the phosphorylated protein levels relative to non-phosphorylated protein levels are shown as a bar graph. As suggested by the reviewer, western blot analysis of FZD6_1CRD, FZD3_1CRD, and FZD3_1linker constructs was performed (Figure 6a). FZD6_1CRD and FZD3_1CRD failed to activate noncanonical signaling, whereas FZD3_1linker and

FZD6_1linker showed increased phosphorylation levels of Akt, ERK, and JNK to some extent but less than that of the wild-type counterparts, in response to Wnt5a.

5. Supp Figure 2 – similar to Fig 1b – the data has been normalised to the FZD wildtype – although a better representation would be the raw expression levels of the constructs.

We have updated the figure, as suggested (Supplementary Figure 3).

6. Supp Figure 5 – Control constructs of FZD4 with a ‘non-canonical’ linker and/or CRD combination should be investigated for Dvl recruitment by both IF and BRET if possible. Similar to Supp Figure 6 (below) lower magnification images with enough cells for analysis should be taken, presented (in combination with the high magnification images) and quantified.

We thank the reviewer for the extensive review and suggestions. We performed both IF and BRET as suggested and increased the number of tested constructs: FZD6 was included as a negative control and FZD4 Δ linker, FZD4_3linker, FZD4_5linker, and FZD4_6linker were tested to observe the effect of the linker. Supplementary Figure 5 of the original manuscript has been revised as Supplementary Figures 12 and 13 (high and low magnification IF images, respectively) in this revised manuscript. The recruitment of Dvl2 was quantified by counting the cells that showed ‘none’, ‘partial’, and ‘complete’ recruitment (representative images are shown in Figure 4d), and the results are plotted as a bar graph (Figure 4d).

We have also included the BRET data as a bar graph in Figure 4c.

7. Supp Figure 6 – the IF is not very convincing – few numbers of cells look like they are actually expressing the constructs – 1 cell has been picked out of the entire image – there is no quantification of the co-localisation and to my eye the colocalization of Fzd6_1linker +Wnt5a looks to be stronger than that of Fzd6 + Wnt5a itself. Lower magnification images with enough cells for analysis should be taken, presented (in combination with the high res) and quantified.

We thank the reviewer for this suggestion. We have captured IF images of both high and low magnification, and the images are presented in Supplementary Figures 20 and 21. Quantification was done as described above (#6), and the plotted graph is shown in Figure 6e.

Specific Comments on Materials and Methods:

1. Generation of an FZD 1, 2, 4, 5, 6, 7, 8 null HEK293T cell line: Can the authors provide evidence of the generation of a FZD6 $^{-/-}$ cell line by western blot, or FACS analysis?

We thank the reviewer for pointing out the missing evidence for the validation of the generated cell line. However, we redid all the cell assays with dFZD1-10 $^{-/-}$ HEK293T cells (Eubelen et al., 2018) as suggested by another reviewer and therefore, we believe that the western blot in question is no longer necessary.

2. Construct design – the authors should include primer sequences and method of cloning.

We thank the reviewer for the suggestion. We realized that our description of FZD construct design was not as clear as it should be and have created Supplementary Table 1 showing all the information of the constructs and Supplementary Table 2 showing the primer sequences used for cloning. We have also updated the methods section to include more information on the cloning method.

3. TOPflash luciferase assay – why was the media changed to growth media and Wnt1/Wnt3a were co-transfected – would it have not been better to serum starve and add recombinant Wnt1/Wnt3a – this would have been better to correct for transfection efficiency and ensure that the signal seen was primarily driven by the adding of Wnts – full growth serum can result in high levels of background TOPflash signal.

We thank the reviewer for the suggestion. Wnt ligands are well-known for being difficult to purify. While we used recombinant Wnt3a, we could not find a credible source for recombinant Wnt1, besides one product that was refolded from *E. coli*. We presume that testing with refolded Wnt1 would cause more uncertainty. In order to validate our results, we have compared the results for Wnt3a signaling using plasmid transfection and recombinant protein treatment (Figure 5a & 5b) and have also included negative controls (pcDNA for plasmid transfection and afamin for protein treatment) for each construct.

Specific Comments (Minor):

1. Citation of Wnt5a activation of FZD6 is a review (#20) – the authors should cite the original research article.

We thank the reviewer for the suggestion. We have cited the original research article.

p4: *“The former activates canonical signaling in response to Wnt1, Wnt3a, and Wnt5a^{20,21}, and the latter activates noncanonical signaling in response to Wnt5a²² (Kilander et al. 2014).”*

2. Supps Figure 3 – ANOVA and Tukeys were performed but there was no statistical analysis indicated on the graphs

We thank the reviewer for the suggestion. In the revised manuscript, details on statistical analysis have been included in each graph.

3. Should include other studies (no in vitro and computational) on the study of the FZD linker domain. E.g (PMID: 33290484)

We thank the reviewer for the suggestion. We have included previous computational studies on the FZD linker domain in the references.

p3: *“Computational modeling was attempted to understand how the ligand-binding domain is linked to the transmembrane domain of FZD^{10,11} (Bang et al. 2018, Agostino and Öther-Gee Pohl 2020).”*

4. Manuscript should be thoroughly checked for typographical errors – e.g (2 µg ml⁻¹; Clontech) in “Generation of an FZD 1, 2, 4, 5, 6, 7, 8 null HEK293T cell line”

Thank you for pointing this out. This part was removed in this revised manuscript because we performed cell-based assays in a new cell line. We have checked our manuscript thoroughly for typographical errors.

Reviewer #2 (Remarks to the Author):

Seung-bum Ko et al., present a study on the functional role of FZD linker in the WNT signalling. This work can be seen as a continuation of Bang et al., (2018) and Valnohova (2018) where the direct impact of different FZD-linker modifications was studied. The present work could be viewed as a substantial addition to the field if not for many methodological issues that in fact make the results questionable, and prevent the publication of the work in its current form. I will go Figure by Figure in my analysis:

Fig.1 While some colour changes can be done to make the Fig 1A more understandable to a reader, cell surface analysis should be re-done as HEK cells are notoriously bad for flow cytometric analysis (please refrain from using the term “FACS” as no cell sorting has been performed in the manuscript). Did the authors use median fluorescence intensity or mean fluorescence intensity? How did they assure that the cells were alive? Flow plots should be attached. Furthermore, if microscopic analysis of fluorescence was also attempted, then it is not quantitative, and prone to being subjective (authors show one transfected cell in each figure, which is not really convincing). The authors should use e.g. live-cell ELISA to get numbers representative of a whole-cell population in a well/on a dish. If the issue with the expression level of the receptors is not fixed, then it is impossible to understand the subsequent TOPFlash results as they are to some extent proportional to the cell surface expression levels of FZDs.

We thank the reviewer for the valuable suggestions on flow cytometric analysis. We redrew Fig1a hoping to make it more intuitive. We do understand the reviewer’s concern in using HEK cells in flow cytometry; however, as we have performed all our assays using HEK cells, we believe that it is necessary that we perform the expression comparison using HEK cells as well.

We believe that, in the revised manuscript, we have shown a better presentation of flow cytometric analysis of each construct with quantitative analysis (Supplementary Figure 3). We clarified in more detail how we performed flow cytometry in the methods section as well. Gated population was selected based on FSC vs SSC to ensure dead cells were excluded and all the flow plots are shown in Supplementary Figure 3. We have plotted the bar graph with the percentage of cells expressing the target protein (Supplementary Figure. 3c). A Live-cell surface ELISA was also performed for all the constructs, in order to cross validate the surface expression of constructs (Supplementary figure 4).

All our results show that in our overexpression system, there is no significant variation in construct expression.

Fig. 2 The authors need to explain how the “Relative WNT-reporter activity” was calculated. In my opinion, since the authors are interested in WNT-induced effects, they should calculate the ratio of firefly to renilla signal for vehicle- and for WNT-treated and FZD-expressing cells. It is known that different FZDs produce different basal TOPFlash signal so it needs to be clearly stated what the graph represents. Additionally, please use dFZD1-10 HEK293T cells (Eubelen et al., 2018), they should be considered as a gold standard now in the field....What is the reason for creating/using a cell line that is not superior to what is already available?

We are grateful for the extensive review, and we have clarified the calculation in the methods section. We have indeed normalized the activity by dividing the luminescence signal intensity of TCF/LEF luciferase by that of Renilla luciferase. The “Relative WNT-reporter activity” was calculated by setting pcDNA-transfected vehicle-treated group as the reference point of one. We have included a description of the TOPFlash assay method in the Methods section of the revised manuscript.

Following the reviewer’s suggestion, we have redone all assays using dFZD1-10^{-/-} HEK293T cells, with the original author’s permission. The basal activity level for each construct was plotted in all

TOPFlash assay graphs.

Fig. 3 The AP analysis is a nice one but the WNT binding assessment can be substantially improved with NanoBRET. It will allow obtaining quantifiable data. Why didn't the authors use all the constructs they generated and ran AP on them?

We thank the reviewer for the suggestion. However, considering the time and effort required, it would be extremely challenging to do NanoBRET on the current data. We have instead followed the reviewer's suggestion on performing AP staining on all the constructs and have added the data to Figures 3, 5c, and 6b and Supplementary Figures 5 and 16. Also, as suggested by another reviewer, the AP-stained area was quantitatively analyzed using the ImageJ and is displayed as a bar graph for each construct (Anti-FLAG and anti-Wnt3a or anti-Wnt5a for each construct).

Fig. 4 All the BRET data should be shown as titration curves with an increasing acceptor amount with or without WNTs. How do the authors know that the interactions are not bystander, e.g. that they do not appear to have a linear, non-saturable increase on a titration curve? BRET experiments are a beautiful way to study protein-protein interactions but they certainly require proper controls...

We thank the reviewer for the thoughtful suggestion regarding BRET. We have tested different ratios of BRET acceptor/donor and have plotted the Δ BRET graph in Supplementary Figures, 6, 8, 10, and 17 for all the pairs that were used in the experiment.

Fig. 5 Same issues as for Fig. 2 Please show anti-FLAG staining for Fig 5D.

As per the reviewer's suggestion, we have updated the figures with anti-FLAG stain (Figures 3, 5c, and 6b and Supplementary Figures 5 and 16).

Fig. 6A needs more comments in the text. How was the statistical analysis performed? Immunoblots provide only semi-quantitative results....; Fig. 6B needs titration curves. If it happens to be true, then the authors should comment more on the apparent lack of FZD1-DVL proximity. I am not sure how the authors concluded that FZD-DVL belongs to the non-canonical pathway. Additionally, please state that DVL2 was used.

We thank the reviewer for the suggestions. We have made a mention that Dvl2 was used in this study.

(6a) Western blot analysis was performed three times, and one representative data set is presented in Figure 6a. The phosphorylation level was quantified using the ImageJ, and the relative phosphorylated protein level with respect to the non-phosphorylated protein level is shown as a bar graph.

(6b) The titration curve for each pair is shown in Supplementary Figure 17. In the absence of Wnt5a, the BRET signal for the FZD1-Dvl2 pair was similar to that for the other FZD-Dvl2 pairs (Supplementary Figure 19). Fig. 6B of the original manuscript (Fig. 6d in the revised manuscript) showed Wnt5a-induced Dvl2 recruitment. As FZD1 does not respond to Wnt5a, FZD1 did not show Wnt5a-induced Dvl2 recruitment.

It has been reported that in the noncanonical Wnt signaling pathway, Dvl1/2 plays a key role in governing polarity and cytoskeletal rearrangements of a cell in response to Wnt5a (reviewed by Wallingford and Habas. *Development* (2005) 132 (20): 4421–4436; Sinha et al. *Dev. Cell* (2012) 370:135-44). Because it has been known that Wnt5a binding to FZD6 activates noncanonical Wnt signaling and we observed that Wnt5a induced FZD6-Dvl2 proximity, we believe that this BRET data suggests that Dvl2 is recruited to FZD3/6 in the non-canonical pathway upon Wnt5a binding to FZD3/6, although we do not know the exact molecular details of the downstream non-canonical

signaling.

Supplementary figures: the issues are recurring; proper quantification of expression levels is crucial.

We thank the reviewer for the suggestion. The surface expression of each construct was quantified with flow cytometric analysis and surface ELISA.

Other issues:

- Please use FZD not FZD receptors. FZDs are WNT-receptors.

Thank you for pointing this out. We have corrected the error.

- Please explicitly specify the number of technical replicates and independent experiments from which data are shown in the figures (for each figure).

Thank you for pointing this out. We have included the number of replicates for each figure.

The authors should also comment on:

- FZD3 and FZD6 WNT-induced TOPFlash signaling upon CRD+linker swap (Fig 5 A and B). Assuming the TOPFlash signal calculations are fine (or will be fixed and still shown the same trend), is it really a meaningful increase given the fact that FZD4 control increases the WNT signal about 60x (WNT-1) to ca. 120x (WNT-3A)? Anyhow, if FZD3 and FZD6 core can indeed signal via B-catenin dependent pathway upon ligand binding to a CRD-linker of another FZD, that will probably be the single most important finding of the paper, and one that will have an impact on the field.

We appreciate the reviewer's comment. We do realize that the signaling power of FZD3_4CRDlinker and FZD6_4CRDlinker is extremely small compared to that of Fzd4 WT; however, the signaling activity of FZD3/_4CRDlinker was repeatedly observed to be statistically significant, and we would like to note that given that FZD3/6 are strictly noncanonical Wnt receptors, this is no small feat, especially when compared to the absence of a signal from FZD3/6_4CRD. In addition, during revision we discovered that when we treated the cells with purified recombinant Wnt3a instead of co-transfection with the Wnt3a DNA, signaling difference between FZD3/6_4CRDlinker and FZD4 became less drastic, as demonstrated in Figures 5a and 5b. We believe that this difference is caused by a longer and more excessive exposure to Wnt3a when DNA was co-transfected.

Together with our AP-staining data showing Wnt3a binding to FZD3/6_4CRDlinker, these data suggest that there may be other key factors besides ligand binding to the extracellular region of FZD to efficiently transmit the signal into the cytoplasm. While we do not have a clear explanation at this point, we do believe that this is an interesting piece of data in understanding the molecular basis of Wnt signaling activation.

We discussed about this issue in p10-11 as below.

p10 : *"In particular, when recombinant Wnt3a protein was treated, the difference in signaling activity between FZD3/6_4CRDlinker and FZD4 became less drastic (~3-fold), as the exposure to Wnt3a was more controlled (Fig. 5b). It should be noted that the FZD6_4CRD mutant failed to elicit any response, again highlighting the importance of both CRD and the linker in signaling."*

p11: *"Therefore, our results suggest that the swapping of the CRD and linker may, to a certain extent, direct FZD towards a signal response that is not naturally inherent in that FZD subtype by conferring ligand-binding ability, but other parts besides the extracellular domain may also be required for efficient signal transduction into the cytoplasm."*

- expression of their chimeric receptors (FZD4_6CRDlinker and FZD6_4CRDlinker) in comparison to the ones published by Kozielwicz et al., 2021. The reviewer realizes obviously that here different fusing points were used (several amino acids of the linker left with the receptor core in the present study).

Thank you for pointing this out. As the reviewer pointed out, the CRDlinker chimeras used by Kozielwicz et al. are different from ours in that their CRDlinker domain was designed to contain all three conserved Cys residues in the linker domain.

We include this information in the Discussion section as below (p14).

p14 : *“However, it is noteworthy that the CRDlinker chimeras used in their study are different from ours in that their CRDlinker domain was designed to contain all three conserved Cys residues in the linker domain.”*

My other comments:

- confocal images of DVL recruitment serve as a good addition to the BRET data.

We have included the IF images of DVL recruitment in Figure 4d and Supplementary Figures 12, 13, 20, and 21.

Reviewer #3 (Remarks to the Author):

The authors have previously shown that the linker domain of Frizzled4 plays an important role in Norrin binding and signaling. This discovery was likely to lead them to investigate the role of the linker domain in the canonical and non-canonical Wnt signalings and in the molecular interactions that underlie them. Basically, their approach is unique and could provide new insights into the mechanisms underlying diversification of canonical and non-canonical Wnt signaling.

However, I have two major concerns with this paper. One is in their interpretation of BRET analysis. The other is about Frizzled that was used for comparison in the experiment shown in Figure 6.

First, I feel there is a problem with the interpretation of the BRET results shown in Figure 4a,b,c. I agree that there is a significant difference in delta BRET between the positive FZD4 and FZD4_5linkers and the negative ones containing FZD4_3linker and FZD4_6linker in the presence of Wnt3a. However, the delta BRET scores are also reduced in those positive for Wnt3a expression. Therefore, it is difficult to conclude from these data the molecular interactions between FZD4 and its partners as described by the authors. Strictly speaking, the oligomerization of FZD4 and FZD4_5linker and their interactions with LRP6 and Dvl are reduced by Wnt3a expression, but the reduction is smaller than that of FZD4_3linker and FZD4_6linker. If the authors want to claim that some specific linker domains actually affect ligand-induced FZD oligomerization, as well as ligand-induced FZD interaction with LRP6 and DVL, they should show evidence that these interactions are actually induced by ligands in their system.

We are grateful to the reviewer for the suggestion. We realize that our presentation of data is not easy to follow, so we have reorganized the figures, in addition to updating the figure legends in order to clarify our results. The updated graphs show FZD4 and FZD4_5linker with clearly higher value of Δ BRET for homo-oligomerization, LRP6 and Dvl2 recruitment compared to M2R and FZD6, indicating enhancement in each upon Wnt3a treatment (Figure 4a, b, c). FZD4 Δ linker, FZD4_3linker and FZD4_6linker show some signals but their signals are significantly lower than those of FZD4 and FZD4_5linker.

We also provide BRET data in the absence of Wnt3a (Supplementary Figures 7, 9, 11).

Second, if we want to properly investigate the influence of linkers in the non-canonical Wnt pathway, it is highly desirable to compare the linker domains of FZD6 with those of FZD3, 4, and 6. I feel that this part is too preliminary and needs significant improvement if the authors want to include this part in the same paper.

We agree with the reviewer that our manuscript needs additional data to support the hypothesis of linker impacting the noncanonical signaling. Since FZD4 has been reported to bind Wnt5a and induce canonical signal (Mikels and Nusse, 2006, doi:10.1371/journal.pbio.0040115), we used CRD/linker of FZD1, instead of FZD4, to make CRD/linker swap constructs as non-responsive substitute to Wnt5a. In updated Figure 6a and Supplementary Figure 16, we have included both FZD3 and FZDF6 in our western blot analyses and AP staining assays.

In addition, we only described what we actually observed instead of generalizing the effect of linker on noncanonical signaling.

p12: *“Altogether, the linker domain affects FZD homo-oligomerization and Dvl2 recruitment of the noncanonical FZD subtypes, FZD3 and FZD6, in the presence of Wnt5a. Since we showed only the changes in the phosphorylation levels of ERK, JNK, and AKT upon Wnt5a treatment, a more systematic analysis is required to demonstrate the essentiality of the linker domain for noncanonical Wnt signaling in general.”*

In addition, their statement in Abstract, “Conversely, a FZD6 chimera, containing the FZD4 CRD and linker, binds to Wnt3a with similar affinity as FZD4 but does not produce a full canonical signal”, may mislead readers. The point to be asserted here seems that “the FZD6 chimera, which contains the CRD and linker of FZD4, enhances Wnt canonical activity, ligand-dependent oligomerization, and interaction with LRP6 more than FZD6”.

Thank you for pointing this out. The message that we intended to convey is given below.

FZD6 chimera, containing the FZD4 CRDlinker, binds to Wnt3a as FZD4 does, suggesting that CRDlinker is a ligand binding site. However, our TOPFlash assay showed that this FZD6 chimera does not have full canonical activity, although it exhibited canonical activity above the basal level, proposing that the Wnt3a-CRDlinker interaction is not the only factor involved in transmitting the Wnt3a-mediated canonical signal into the cytoplasm.

To deliver our message clearly, we wrote the statement as below in the main text.

p11: *“Therefore, our results suggest that the swapping of the CRD and linker may, to a certain extent, direct FZD towards a signal response that is not naturally inherent in that FZD subtype by conferring ligand-binding ability, but other parts besides the extracellular domain may also be required for efficient signal transduction into the cytoplasm.”*

In abstract, we modified the sentence as below.

p2: *“A similar effect was observed for noncanonical signaling. A FZD6 chimera containing the FZD1 linker showed reduced Wnt5a binding and impaired signaling in ERK, JNK, and AKT mediated pathways.”*

Specific points

Page 3, last sentence; The Wnt-Frizzled interaction has been studied by several groups. We recommend that you cite these papers as REFERENCE here. Wnt-Frizzled interaction has been investigated by several groups (Bhanot et al, 1996 ; Wu and Nusse, 2002; Takada et al, 2005).

We thank the reviewer for pointing this out. We have added the references.

Figure 1b, Supplementary Figure2b

It is unclear how the relative surface expression of each Fzd protein is measured: is the percentage of cells with Fzd protein on the surface relative to the entire transfected cell population, relative to the entire cell population expressing Fzd, or something else? What is the procedure for transfection and FACS analysis in these experiments?

We thank the reviewer for the suggestion. We have majorly updated Figure 1b and Supplementary Figure 2. In the revised manuscript, flow cytometry histograms of FZD-expressing cells are included in Supplementary Figure 3. These histograms show the percentage of cells expressing the construct. In addition, we performed surface ELISA for each construct (Supplementary Figure 4).

We have included the methods used for transfection and flow cytometry analyses in the methods section.

Figure 2a,b; Authors should include data showing the Wnt activity of each Fzd-expressing cell in the absence of the Wnt ligand.

We appreciate the reviewer’s thoughtful advice and have made the suggested revision (Figures 2a, 2b).

Page6, line13-14 “FzD4 and FZD5 responded to Wnt1 and Wnt3a, while FZD3 and FZD6 remained silent” : Since Figure 2b shows that FZD3 and FZD6 have higher Wnt activity than the negative control, pcDNA, the authors need to be careful in describing this result.

We thank the reviewer for the suggestion. We redid the assay with the basal level indicated for each construct. Each compared to the negative control, FZD3 and FZD6 do not seem to elicit significant signaling in response to Wnt1 or Wnt3a.

Page6, line15-16 “Replacing the FZD4 CRD with the CRD of FZD3 or FZD6 was shown to completely eliminate the activity of Wnt1 and Wnt3a”; Since Figure 2a shows that FZD4_3CRD and FZD4_6CRD have higher Wnt activity than the negative control, the phrase "completely eliminate" is not appropriate in this case.

We thank the reviewer for pointing this out. During reviewing process, we have redid all our analyses using dFZD1-10^{-/-} HEK293T cells in order to eliminate any contamination from endogenous FZD and it has altered our results. We do not observe canonical signal by CRD swap mutants as shown in Figure 2a.

Page 6, line 14; The data display for this statement is missing.

We thank the reviewer for pointing this out. We have included “Figure 2a” in this statement as below.

p6: *“TOPFlash assays showed that FZD4 and FZD5 responded to Wnt1 and Wnt3a, while FZD3 and FZD6 remained silent, which was as expected (Fig. 2a).”*

Page 7, line15; The detection of the interaction between Wnt and Frizzled by immunostaining was already reported in a paper that identified Frizzled as the Wnt receptor (Bhanot et al, 1996).

We have rephrased the sentence as per suggestion and have cited the reference.

Figure 5d; FLAG staining is missing

Thank you for the suggestion. We have added the panel for anti-FLAG staining.

Figure 5e,f; BASAL (without Wnt3a) is missing

Thank you for the suggestion. We performed BRET assays without Wnt3a treatment and have presented the data in Supplementary Figures 9 and 11.

Figure 6; Quantification results of Western blotting should be indicated.

We thank the reviewer for the suggestion. We quantitatively analyzed the western blot data using the ImageJ (Figure 6a).

Figure 6b; BASAL (without Wnt3a) is missing

We thank the reviewer for the suggestion. We performed BRET assays without Wnt5a treatment and have presented the data in Supplementary Figures 18 and 19.

Reviewers' comments:

Reviewer #1 (Remarks to the Author):

The authors have done a commendable job responding to my comments. I am happy with the experimental results which have made the conclusions more robust.

My only query is that that quantification for the western blots in 6a) appear to only be from one repeat. If this is the case the experiment should be repeated to ensure the result is solid.

Reviewer #2 (Remarks to the Author):

In this revised manuscript the authors have addressed a vast majority of my initial concerns. I have only one comment left: please adhere to protein nomenclature guidelines and use uppercase and subscript when needed. Otherwise, I think that the manuscript in its current form is suitable for publication in Comms Bio.

Reviewer #3 (Remarks to the Author):

I understand that the author has responded appropriately to all the points I made earlier.

Finally, I would suggest that the author consider the following point, if possible.

The term "delta BRET" is used both to indicate the difference in the BRET ratio with and without Wnt3a (Figure 4a,b,c, Supplemental Fig. 6, 8, 10) and to indicate the difference in the BRET ratio with the control M2R (Supplemental Fig 7). I think it would be more helpful to the readers if the authors could clearly show this difference on the graphs. For example, in Supplemental Fig. 7, the M2R data could be added to the graph, and the vertical axis could be expressed as BRET ratio.

We greatly appreciate the time and efforts that the reviewers invested in reviewing our manuscript.

As detailed in our point-by-point responses given below (in blue font), we have addressed each of the points raised by the reviewers. In addition, we converted all bar graphs to dot-plot format to show data distribution (Figures 2-6, Supplementary Figures 3c, 4, 5, 7, 9, 11, 14, 15, 17, 19, 20).

Reviewers' comments:

Reviewer #1 (Remarks to the Author):

The authors have done a commendable job responding to my comments. I am happy with the experimental results which have made the conclusions more robust.

My only query is that that quantification for the western blots in 6a) appear to only be from one repeat. If this is the case the experiment should be repeated to ensure the result is solid.

We have shown one representative western blot (Fig. 6a) out of three independent repeats (Supplementary Fig. 16) and the error bars in Figure 6(a) are SEM of those repeats as well. We have clarified that in the methods and legends.

Reviewer #2 (Remarks to the Author):

In this revised manuscript the authors have addressed a vast majority of my initial concerns. I have only one comment left: please adhere to protein nomenclature guidelines and use uppercase and subscript when needed. Otherwise, I think that the manuscript in its current form is suitable for publication in Comms Bio.

As suggested by the reviewer, we have used uppercase for protein names, such as FZD4, WNT3A, and DVL2 according to protein nomenclature guidelines. (In the entire manuscript, we changed Wnt1, Wnt3a, Wnt5a, and Dvl2 to WNT1, WNT3A, WNT5A, and DVL2, respectively.)

Receptor subtypes are usually designated by means of a subscript numeral, but for Frizzled receptor subtypes, numbers without subscripts are more commonly used, such as FZD4^{1,2}.

¹Yang et al., *Nature* (2018) 560:666–670.

“Here we present an atomic-resolution structure of the human Frizzled 4 receptor (FZD4) transmembrane domain in the absence of a bound ligand.”

²Nile et al., *Proc. Natl. Acad. Sci. USA* (2017) 114:4147–4152.

“Here, we determined a crystal structure of human FZD7 CRD unexpectedly bound to a 24-carbon fatty acid.”

In particular, in order to clearly indicate the various subtype chimeras, we used non-subscript numbers for Frizzled subtypes, such as FZD4_5linker.

Reviewer #3 (Remarks to the Author):

I understand that the author has responded appropriately to all the points I made earlier.

Finally, I would suggest that the author consider the following point, if possible.

The term "delta BRET" is used both to indicate the difference in the BRET ratio with and without Wnt3a (Figure 4a,b,c, Supplemental Fig. 6, 8, 10) and to indicate the difference in the BRET ratio with the control M2R (Supplemental Fig 7). I think it would be more helpful to the readers if the authors could clearly show this difference on the graphs. For example, in Supplemental Fig. 7, the M2R data could be added to the graph, and the vertical axis could be expressed as BRET ratio.

We thank the reviewer for the suggestion. We have included the M2R data and displayed them as BRET ratios in Supplementary Fig. 7, 15, and 19 as suggested. In addition, in order to clearly indicate that Fig. 4a, b, c, and Supplementary Fig. 6, 8, 10 represent the differences in the BRET ratios with and without WNT3A, we wrote " \pm WNT3A" in each figure. Similarly, we indicated " \pm WNT5A" in Supplementary Fig. 18.